



# Enhanced flood hazard assessment beyond decadal climate cycles based on centennial historical data

Gerardo Benito[1], Olegario Castillo[2], Juan A. Ballesteros-Cánovas[3], Maria Machado[1], Mariano Barriendos[4]

[1]National Museum of Natural Sciences, MNCN-CSIC, C/ Serrano 115bis, 28006, Madrid, Spain
[2]Dpt. Ingeniería Industrial e Ingeniería Civil, Escuela Politécnica Superior, Universidad de Cádiz, 11202 Algeciras, Cádiz, Spain
[3]Climatic Change Impacts and Risks in the Anthropocene (C-CIA), Institute for Environmental Sciences, University of Geneva, Geneva, Switzerland
[4]Dpt. d'Història i Arqueologia, Universitat de Barcelona. Montalegre 6, 08001 Barcelona, Spain

*Correspondence to*: Gerardo Benito (benito@mncn.csic.es)

**Abstract.** Current climate modelling frameworks present significant uncertainties when it comes to quantifying flood quantiles in the context of climate change, calling for new information and strategies in hazard assessments. Here, state-of-

the-art methods on hydraulic and statistical modelling are applied to historical and contemporaneous flood records to evaluate flood hazards beyond natural climate cycles. A comprehensive flood record of the Duero River in Zamora (Spain) was compiled from documentary sources, early water-level readings and continuous gauge records spanning the last 500 years. Documentary evidence of flood events includes minute books (municipal and ecclesiastic), narrative descriptions, epigraphic marks, newspapers and technical reports. We identified 69 flood events over the period 1250 to 1871, of which,

15 were classified as catastrophic floods, 16 as extraordinary floods, and 38 as ordinary floods. Subsequently, a 2D-hydraulic model was implemented to relate flood stages (flood marks and inundated areas) into discharges. The historical flood records show the largest floods over the last 500 years occurred in 1860 (3450 m$^3$/s), 1597 (3200 m$^3$/s), and 1739 (2700 m$^3$/s). Moreover, at least 24 floods exceeded the perception threshold of 1900 m$^3$/s during the period (1500-1871). Annual maximum flood records were completed with gauged water-level readings (PRE: 1872-1919) and systematic gauge

records (SYS: 1920-2018). The flood frequency analyses were based on (1) Expected Moments Algorithm (EMA) and (2) Maximum Likelihood Estimator (MLE) method, using five datasets with different temporal frameworks (HISTO: 1511-2018, PRE-SYS: 1872-2018, ALLSYS: 1920-2018, SYS1: 1920-1969, and SYS2: 1970-2018). The most consistent results were obtained using the HISTO dataset, even for high quantiles (0.001% AEP). PRE-SYS was robust for the 1% AEP flood with increasing uncertainty in the 0.2% AEP or 500-year flood, and ALLSYS results were uncertain in the 1% and 0.2%

AEP floods. Since the 1970s, the frequency of extraordinary floods (>1900 m$^3$/s) declined, although floods on the range of the historical perception threshold occurred in 2001 (2075 m$^3$/s) and 2013 (1654 m$^3$/s). Even if the future remains uncertain, this bottom-up approach addresses flood hazards under climate variability providing real and certain flood discharges. Our results can provide a guide on low-regret adaptation decisions and improve public perception of extreme flooding.



## 1 Introduction

There are major challenges in dealing with flood hazards on a global scale (UNISDR, 2015). Climate warming is part of the problem, but the challenges will continue to be great as a result of population growth and human occupation of flood risk zones (Kundzewicz et al., 2014). Most "top-down" approaches based on downscaling of nested climate models with hydrological approaches (deterministic and statistical) typically produce uncertain results, with a wide range of scenarios which are difficult to implement on a local scale (García et al., 2014; IPCC, 2012). Given the uncertainty of climate model

projections, a new focus on other risk variables (demography, land use, urbanisation) has led to supporting decision-making processes (Döll et al., 2015). This "bottom-up" approach is based on reducing exposure and vulnerability, but still fail to solve the probability assessment of flood hazards due to the stochastic nature of weather (Kundzewicz et al., 2010). In this regard, the information from past flood events has become an important data source to quantify the links between the occurrence of extreme events and natural climate variability that provide expectations of future climate change.

Over recent decades, paleoflood and documentary flood archives have been used to quantify flood discharge and frequency over centennial-to-millennial time scales with applications for engineering design and risk estimation (Baker, 2008; Benito et al., 2020). A recent pan-European historical archive analysis has identified nine flood-rich periods (30-40 year interval) over the last 500 years; all except the last one (1992-2016) occurring during intervals of colder air temperatures than the interflood period before and after (Blöschl et al., 2020). The existence of flood-rich periods was also demonstrated at

millennial time scales for the Mediterranean basin and at a European scale using paleoflood sedimentary evidence (Benito et al., 2015c). These studies reveal that flood-producing mechanisms, on a local to regional scale, are not necessarily driven by temperature anomalies, but are controlled by the behaviour process of ocean and atmospheric circulation (Woollings et al., 2010; Ballesteros-Cánovas et al., 2019). Thus, long-term flood registers, produced by multiple atmospheric-ocean interactions, contain a wide range of weather anomalies to advise on expected flood extremes in a changing climate and,

more importantly, it reveals what flood dimension is actually possible on a local scale (Baker, 2008).

Historical and paleoflood hydrology have demonstrated their potential to determine the quantitative information of specific flood events over centennial to millennial time spans (Benito et al., 2015a; Cœur and Lang, 2008; Wilhelm et al., 2019). Major advances are related to (1) digitalisation of archival sources, facilitating the search and screening tasks, (2) LiDAR data used for digital terrain models, buildings and urban spaces, (3) application of sophisticated two- or even three-

dimensional hydraulic modelling to estimate peak discharges, local flow velocity and depth, and (4) new statistical procedures and software to feed historical and palaeoflood into quantitative frequency analysis (Benito et al., 2020). Although in the past the use of historical flood archives was mainly considered in academic research circles, recent publication by the US federal government guidelines (England Jr. et al., 2019) shows a growing interest to harness past flood information to evaluate flood hazards and risks (St. George et al., 2020). In the European Union, several countries have

proposed the use of past flood data on the probability of future floods, as part of low-regret actions to solve the uncertainty of downscaling climate model results on a local scale (García et al., 2014; European Union, 2021). Going forward, technical



guidance and new protocols on the use of past flood archives are needed to reach engineers and stakeholders, so it can become a far-reaching and smart tool to cope with flooding (Excimap, 2007; England Jr. et al., 2019).

In this paper, we firstly implement a holistic methodology combining historical flood evidence and a two-dimensional hydraulic model to reconstruct centennial registers of peak discharges and their relationship with natural climate variability and triggers. Secondly, we apply two flood frequency statistical methods to different type and series length of flood data sources (historic, pre-instrumental and gauged records) to evaluate the robustness of flood discharges for low probability events. Thirdly, we consider the historical flood imprint on local communities as a basic tool to improve public risk awareness on urban space, heritage buildings and cultural landscapes. Our results based on this bottom-up approach show a direct guide on flood possibilities beyond decadal climate cycles that can be used to provide a portfolio of low-regret solutions suitable for climate change adaptation.

## 2 Study area

### 2.1 Geographical and physical setting

The Duero river drains the northern Spanish Plateau and flows east-west into the Atlantic Ocean at Porto (Portugal) (Fig. 1a). It is one of the longest rivers of the Iberian Peninsula (897 km) and the largest in catchment area (98,073 km$^2$), of which 78,859 km$^2$ are in Spain and 19,214 km$^2$ in Portugal. The flood records studied are located in Zamora, in the lower part of the Spanish Duero Basin (Fig. 1b), draining a catchment area of 46,137 km$^2$.

The Duero basin is surrounded by the Cantabrian Mountains to the North, the Iberian Range to the East, and the Central Range (Gredos and Guadarrama Mountains) to the South. Geologically, the Duero catchment area comprises two major zones: 1) the eastern side covers the Cenozoic endorheic continental basin (~ 50,000 km$^2$) composed of detritic, carbonate and evaporitic units (Alonso-Zarza et al., 2002), overlain by Quaternary alluvial fans and fluvial staircase terraces developed by fluvial dissection related to the onset of exoreic basin conditions (Martin Serrano, 1991; Rodríguez-Rodríguez et al., 2020); and 2) the west is composed by Palaeozoic granitic and a metamorphic basement, where the fluvial network is deeply incised forming confined river valleys.

The Duero River east of Zamora flows along a 2-3 km wide floodplain (Fig. 1c). In Zamora the floodplain is asymmetric with a 300 m wide floodplain at the southern margin whereas in the northern side the channel is cut on silicified sandstone and conglomerates dating back to Early Cretaceous-Palaeocene (Fm. Areniscas de Salamanca; Delgado-Iglesias and Alonso-Gavilán, 2008). West of Zamora, the river is incised in granite and metamorphic rocks of Palaeozoic age, forming a confined bedrock canyon, with punctuated valley expansions. The combination of a narrow and steep valley floor has been optimal for the development of hydroelectric dam facilities along the Arribes del Duero (narrows of the Duero), an impressive 800 m deep bedrock canyon formed by the Duero at the Spanish-Portuguese border.



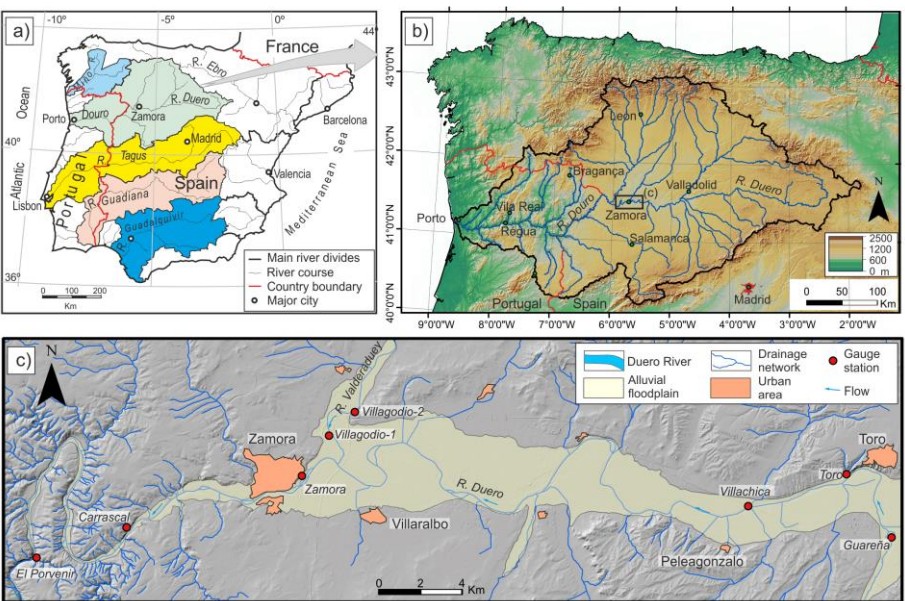

**Figure 1: a) The Duero River Basin (green) in the Iberian Peninsula together with Miño (light blue), Tagus (yellow), Guadiana (orange) and Guadalquivir (dark blue) catchments draining towards the Atlantic Ocean; b) Relief map of NW Iberia with the Duero catchment boundary, main drainage network and major cities. The rectangle at Zamora shows the location of (c); c) Hillshade map with extension of the floodplain between Toro and Zamora. Map shows location of gauge stations in the Duero and Valderaduey rivers.**

## 2.2 Historical urban development and flood documentation

The old city of Zamora, still partially walled, is located on a prominent bedrock hill on the right side of the Río Duero, and treasures a rich architectural ensemble formed by the twenty-four Romanesque churches and monasteries (10th-13th Centuries). The urban expansion towards the east, parallel to the riverside area, took place mainly in three phases during the 11th, 12th and 14th Centuries (Fig. 2a), coinciding with periods of economic and population growth (Gutiérrez González, 1993; Larrén, 1999).

The five oldest Romanesque churches located on the Duero River were built during the late 11th and 12th Centuries (Santo Tomé, Santiago el Viejo, San Claudio de Olivares, San Cebrián and Santa Maria la Nueva, Fig. 2a). Other Romanesque churches bearing marks corresponding to historic flooding were built during the first half of 13th Century (Santa Maria de la Horta, San Frontis, Dueñas de Cabañales, San Leonardo, and Santa Lucia). The first known bridge (Puente Viejo), located near the Olivares mill, was destroyed by the 1310 flood event (Marquina, 1949b) though several basal piers remain visible (site 24, Fig. 2b). The medieval Stone Bridge, still in use, was finished during the 13th Century although with a first written reference in 1167 as "Ponten Novum" or New Bridge (Enríquez-De-Salamanca, 1998; site 8, Fig. 2).





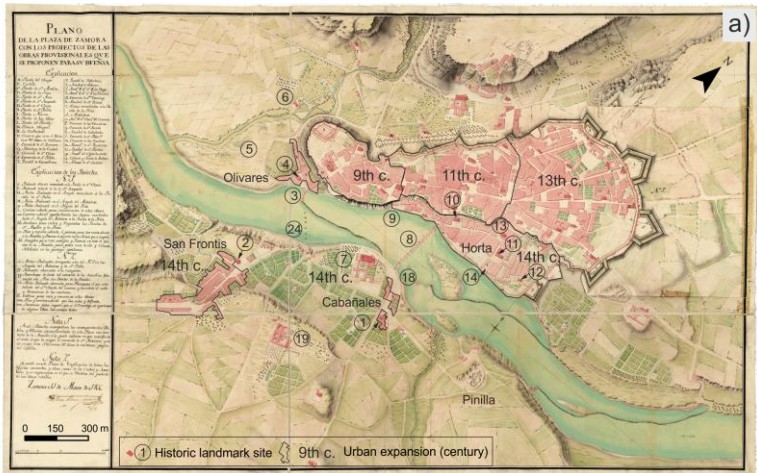

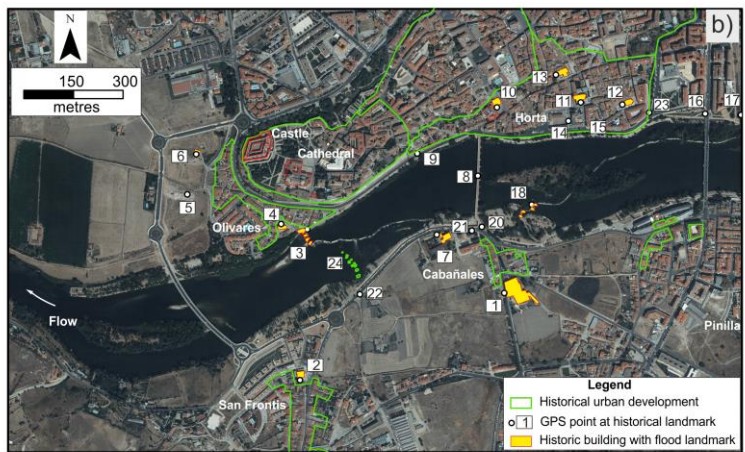

**Figure 2: Historic flood landmarks and GPS point locations. a) Historic map of Zamora (1:2900 in scale) published on March 11, 1766 by Juan Martín Cermeño. Old graphic scale of 200 toess (1toe = 13.5 cm) Black polygons illustrate the timing of urban development through time in the old city (9-13th centuries) and suburbs (14th century and beyond). The city of Zamora was**
120 **rounded by walls that in the old city had defence purposes and in the suburbs protected against flooding; b) Ortophoto of Zamora (year 2017) showing same urban expansion areas as a). Legend: 1) Las Dueñas convent; 2) San Frontis church; 3) Olivares mills; 4) San Claudio de Olivares church; 5) Campo de la Verdad (likely location of old Santa Clara convent); 6) Santiago de los Caballeros church; 7) Old San Francisco convent; 8) Stone bridge; 9) Gate of Pescado dated at 14th century (a new one was built in 1849; Fig. 4b); 10) Santa Lucia church; 11) Santa Maria de la Horta; 12) Santo Tomé church; 13) San Leonardo; 14) Cavalry**
125 **headquarters; 15) Mengue Avenue; 16) Iron bridge; 17) Railway bridge; 18) Cabañales mills; 19) San Jerónimo; 20) Survey monument N.P. 1482 (IGN 1925); 21) IGN Survey mark 305004; 22) IGN Survey mark 305005; 23) IGN Survey mark 305003. Source: a) Biblioteca virtual Ministerio de Defensa https://bibliotecavirtual.defensa.gob.es/; b) National Geographical Institute.**

The number of reported floods increases in parallel with the demographic growth during the 14th Century that brought the

third major urban expansion and new commercial and artisanal activities (Fig. 2a). These activities were carried out in the

130 new suburbs next to the Stone Bridge (Horta neighbourhood), as well as in the areas surrounding the three water mills. The

Olivares mill and its neighbourhood sit on the right bank, the activity of which, linked to the wool, cloth and tannery

industry, was already referred to at the end of the 11[th] Century (Gutiérrez González, 1993). The water mills of La Pinilla





(12[th] Century) and Cabañales (15[th]-16[th] Centuries) on the left bank, traditionally an area of meadows, were known for their tannery and pottery manufacture. A later economic expansion took place during the 18[th] Century under the protection of the Spanish Crown.

The flood references between the 14[th] and 18[th] Centuries are concentrated in the riverside areas of new commercial and artisanal expansion to the east of the Stone Bridge (in the streets of San Julian, La Plata-Balborraz, Baños, Horta, Cuartel de abajo). The frequent flood references are linked not only to its geomorphological and hydraulic settings, but to a greater exposure and vulnerability. On the left river margin, flood reports are related to ecclesiastic buildings (Las Dueñas, San Francisco, San Frontis) and their surrounding orchards.

The urban development that took place from the 19[th] Century onwards is also reflected in the increase and spatial distribution of buildings and infrastructures associated to flood damages. At the beginning of the 19[th] Century, with the French military invasions, large parts of the city walls were demolished (eastern walls in Fig. 2a), which still offered not only military protection (Larrén, 1999), but also protection against flood risk in part of the city of Zamora (south and east of Horta neighbourhood). However, the greatest change took place in the mid and late 19[th] Century, with the modernisation of the city (Segundo Viloria's project in 1880), as well as in the mid-20[th] Century, with the construction of the Vigo road along the Duero River right margin, from Olivares to the confluence with the Valderaduey River (Fig. 2b, connecting sites 9, 15, 16, 17). In this urban growth, a significant portion of the remaining city walls were removed or incorporated within new buildings (Gutiérrez González, 1993). This city expansion brought a narrowing of the river section at the floodplain, and changes in the location and growth of fluvial bars, mainly next to the bridges and weirs. Furthermore, the limited natural space of the riverside area was highly transformed during the construction of the sewage system and two new bridges (1) the old railway bridge built in 1895, reformed in 1933 and without traffic since 1986, and (2) the "Iron Bridge" (1892-1900) still in use for road traffic circulation (Fig. 2b site 16). Other infrastructures are diversion weirs related to watermills historically used for grinding wheat (Fig. 2). The most recent bridge was opened in 2013 connecting the suburbs of San Frontis and Olivares.

## 2.3 Climate and flood hydrology characteristics

The climate is continental Mediterranean in most of the catchment area, with a strong temperate oceanic influence towards the mouth of the Duero River in the Atlantic Ocean at the city of Porto. Temperature and precipitation regime are characterised by a marked seasonal and monthly variability. Summers are hot and dry, and winters are typically mild and relatively wet. Rainfall is mainly produced by cold Atlantic frontal systems crossing the Iberian Peninsula from November to April. There is a strong west-east rainfall contrast due to the relief effect, with annual rainfall at the lower Duero in Porto of 1175 mm, whereas at the Spanish Duero basin it is 580 mm with a wide inter-annual variability ranging from 350 and 800 mm. Similarly, daily maximum rainfall on the Portuguese side may reach 200 mm, whereas for the Spanish Duero it ranges between 60-100 mm.



In Zamora, the Duero River mean annual discharge is 99 m³/s (period 2002-2017), with flow partially regulated by reservoirs. General hydrological characteristics are: (i) maximum discharge from December to May, (ii) a peak between February and March, and (iii) minimum discharge from July to September. This seasonal pattern is influenced by a mix of snowmelt and rainwater from tributaries draining the Gredos and Cantabrian mountains. Most of the largest floods are related to persistent winter rainfall (several weeks) associated with successive passage of Atlantic fronts, occasionally

combined with snowmelt at mountain ranges surrounding the catchment area. Extreme flood discharges may be 30 times greater than the mean discharge, that yield one of the largest specific peak discharges compared to similar European catchment areas (Pardé, 1953; Benito et al., 2015a). Although the number of hazardous floods has decreased over recent decades, the Duero River Watershed Authority reported for the Zamora Province flood damages of 270 k euros/year over the period 2000-2013.

The Portuguese Douro catchment area is more likely to generate catastrophic floods than the Spanish side due to the convergence of significant tributaries with a much larger runoff contribution (Vehlas, 1997). For instance, the largest gauged flood of the Douro River in Porto exceeded 16,000 m³/s in 1909, whereas this event in Zamora recorded 2155 m³/s. Among the reasons of this flood disparity are a high instability and advection of humid air masses towards Portugal, higher relief decreasing concentration time, and more impervious igneous and metamorphic bedrock. In contrast, the Spanish Duero River

flows on wide valleys, detrital bedrock and well-developed floodplains with a lack of flood peak convergence, since the headwater flows reach the peaks with a delay with respect to those coming from the middle and lower valley. For example, the 1909 flood peak was recorded in Porto on the morning of Dec 24 (from the Portuguese catchment area) whereas at the Spanish side in Valladolid the peak occurred on the afternoon of Dec 25, with the flood wave reaching Zamora during the night. As a result, the ranking of the years in which the largest floods occur in Zamora may differ from that observed in

Régua and Porto at the lower Douro basin.

## 3 Data sources and methods

### 3.1 Instrumental records

The Duero River hydrological gauge stations next to Zamora (Fig. 1c) comprise: (i) Carrascal (Station N.2066; period 1918-2021) located 8 km downstream of Zamora, (ii) Zamora (Station N.2121; 2002-2021), (iii) Villachica (Station N.2096; 1929-

1967), and (iv) Toro (Station N.2062; 2011-2021), the latter two are located at 25 and 28 km upstream of Zamora, respectively.

The oldest gauged records correspond to water-level readings taken on a daily basis in the El Porvenir gauge station, covering the period 1880 and 1943. The gauged section is located c. 23 km downstream of Zamora at the San Roman hydroelectric station (operating since 1903). According to Marquina (1949a), the Porvenir rating curve was well established

to a stage of 5 m (1450 m³/s) using the Villachica gauge data, whereas for higher discharges extrapolation of the curve may result in errors of c. ±10%. The flood record was completed with the Carrascal station (1920-present) managed by the





Iberdrola hydropower company, which allows robust estimations of historic floods in Zamora. The Toro and Villachica stations were used to test the gauged data at Carrascal, particularly those corresponding to flood peaks.

## 3.2 Historical data sources

Our historical flood database was collected from published compilations, unpublished documents, epigraphic marks, historical maps, photos and newspapers (listed in supplementary material). The documentary flood records in Zamora essentially comprise a continuous series from 1545 to 1860, a non-continuous dataset between 1250 and 1545 collected from ecclesiastic and municipal archives. In the Cathedral of Zamora Archives, the "Extracts of ecclesiastical agreements" comprises 2 volumes over the period 1601 to 1745 and the "Books of ecclesiastic agreements" includes 34 volumes over the

period 1601 to 1913. The Municipal books (*Libros de Actas*) comprise 259 volumes over the period 1500-1899, although there are some missing documents over the periods 1503-1507, 1521-1530, 1576-1585. Many references of those books and local chronicles were collected by historiographic collections, namely the Historical Memoirs of the City of Zamora (4 volumes) by Fernández Duro (1882). Climate and hydrological extreme events descriptions with reference to flooding in Zamora were also compiled in local ecclesiastic chronicles, namely by Zatarín-Fernández (1898), and in the geological

description of the Zamora Province by Puig y Larraz (1883). Large floods typically affected historic buildings, such as churches, convents, bridges, walls and gates, the traces of which are recorded in architectural catalogues that describe inscriptions and flood marks, repair work to flood damages or changes in the location of ecclesiastical communities due to the effects of major floods (Gomez-Moreno, 1927; Antón, 1927).

The most outstanding historical flood compilation effort was carried out by Rodríguez Marquina (1941-1949) that analysed

all available historical and hydrological reports on flooding for the construction of the dams in the Duero and Esla rivers. Marquina's manuscripts provided a highly detailed description and height survey of epigraphic marks, including some flood marks that later disappeared during restoration works. Marquina also compiled original water-level gauge readings from the Duero and Esla rivers at different locations (e.g. Porvenir gauge). The temporal evolution of floodplain areas, buildings and riverine structures (weirs, bridges, mills, orchards, etc.) were evaluated using historic maps, drawings and etchings from

historic times, namely by Anton van der Wyngaerde in 1570 (Kagan, 1986; Rodríguez-Méndez and García-Gago, 2014), Josep Auguier in 1756 (Museum of Zamora) and Juan Martín Cermeño in 1766 (Digital Archive of the Ministry of Defence; Fig. 2a). The most detailed historical maps belong to the collection of plans of the municipal architect Segundo Viloria in 1880 (Provincial Historical Archive of Zamora) and Zamora's municipal maps (19th-20th Centuries). A final source of flood data, mainly over the last four decades was obtained from local and regional newspapers (Heraldo de Zamora, La Voz de

Zamora, La Opinión, Impero, among others) and national press (Ahora, La correspondencia de España, ABC, La Vanguardia). Regarding more general reference to the Duero River and its tributaries, an outstanding compilation of flood dates from historical data sources was compiled by Fontana Tarrats (1971-1977).

The reported floods were compared with historical series in the lower Douro River in Portugal (Loureiro, 1904; Aires et al., 2000; Amorim et al., 2017; Alcoforado et al., 2021). Flood severity was classified according to Barriendos and Martín-Vide





(1998) into three flood categories: 1) Ordinary, causing overbank flows of low-moderate intensity and temporary disruption of the human activities; 2) Extraordinary, causing overbank flows of moderate intensity, with limited damages to crops, houses and river dykes, and 3) Catastrophic, causing extensive overflow with significant damage to agriculture, mills, and/or destruction of houses and infrastructures.

**3.3 Two-dimensional hydraulic modelling**

Flood water level (stage) related to historical floods include:(i) landmarks and observed water depth measures at sites reached by the flow (e.g. monastery, bridge, chapel, etc.), (ii) description of flooded areas (e.g. floodplain sectors, orchards, mills), (iii) non-flooded areas, (iv) comparative flood level of subsequent historical floods (e.g. said flood reached lower levels than previous one). The documented flood evidence can be used to estimate exact or relative (minimum or maximum) flood discharges associated with observed water levels. The conversion of flood level to discharge is obtained by matching a

modelled water surface elevation for a given modelled discharge to the surveyed elevation of the known historical flood level (Benito et al., 2020).

Discharge estimation by hydraulic modelling was carried out using a two-dimensional hydraulic model (Iber) which solves the depth averaged shallow water (2D Saint-Venant) equations using a finite volume method with a second order roe scheme (Bladé et al., 2014; www.iberaula.com). This two-dimensional hydraulic model is particularly suitable for flow in alluvial

floodplains with secondary currents. The model uses a non-structured mesh consisting of triangles or quadrilateral elements whose spatial resolution were set to 20 m for the channel bed, 10 m for the channel margins and floodplains, 1 m for river bars, and1 to 5 m in the city streets and at major infrastructures (Fig. S1). The 13 km length modelled reach extends from the Carrascal gauge station to the Duero River junction with the Valderaduey River (at the highway A-11; Fig. S1a). The Carrascal section is situated at a narrow bedrock canyon producing a backwater effect upon the upstream alluvial reach

where the city of Zamora is located. The mesh elements were built from a cloud of LiDAR data with average distance of 1.4 m (some are below 1 m) supplied by the Spanish National Geographic Institute (IGN; https://www.ign.es). The topography of the river channel bottom and banks was obtained from field surveys with echo-sounder device at cross-sections of 40 to 500 m distance (LINDE project, MITECO). The surveyed points from 268 cross-sections were extracted and integrated with the LiDAR cloud data using the Spatial Analyst tools in ArcGIS v.10. The bridges were introduced to model historical flood

discharges considering their construction date.

Manning's n values were assigned from land use map classes (Corine Land Cover map) following the methodological guide of the National Flood Inundation Hazard Map (MMA, 2011). In our study, the initial set of n values was defined in 0.04 for the main channel and between 0.045 and 0.1 for the floodplains. Model calibration was performed using flow discharge-stage records at two-gauge stations, namely the Carrascal one located at the furthest downstream cross-section and Zamora

Station in the upper sector of the modelled reach. For this calibration, the outflow stage for successive increments of inflow discharges in the upper section in Zamora was compared to the rating curve at the Carrascal gauge station. The difference in



stage between the model and the gauge station for a discharge of 1100 m$^3$/s was 2 cm, and for a discharge of 3100 m$^3$/s was 12 cm. After the model calibration, Manning's n at the channel was set at 0.035.

The flood marks and flooded sites mentioned on documentary flood evidence were surveyed using a Trimble GPS, supported
by four geodetic survey monuments of the Spanish National Geographical Institute (Numbers 305003, 305004, 601005 and 601006; Table S1). We modelled successive increments of inlet water discharges at the upstream reach to simulate a steady flow. The hydraulic model provided a peak discharge *vs.* water stage relationship for 30 sites with known historical flood evidence (epigraphic marks, documented heights, description of flooded/non-flooded areas). We used the mean and standard deviation to estimate average peak discharge for each historical flood or used the minimum or maximum flood values
according to the mean of the documentary flood evidence.

### 3.4 Flood frequency analysis

Flood data stationarity for censored samples (historical and systematic flooding) was confirmed using Lang's test (Lang et al., 1999). This test assumes that stationary flood series can be described by a homogeneous or stationary Poisson process. The 95% tolerance interval of the cumulative number of floods above a threshold or censored level is calculated. Stationary
flood series are those that remain within the 95% tolerance interval (Naulet et al., 2005). The flood frequency analysis was performed with two computer programmes: PeakFQ Version 7.2 (Flynn et al., 2006; Veilleux et al., 2014) and AFINS (Botero and Francés, 2006; Botero and Francés, 2010). The PeakFQ program applies a generalised method-of-moments estimator denoted the Expected Moment Algorithm (EMA, Cohn et al., 1997), whereas AFINS uses the maximum likelihood method (Frances, 2004). Flood frequency analysis were carried with different combinations of three datasets (1)
documentary floods using minimum flow and/or peak discharge estimated from landmarks and reported flood descriptions (historic dataset, HISTO); (2) water-level readings on a scale gauge transformed into discharge at Porvenir station (pre-instrumental dataset; PRE); (3) continuous systematic records at Carracal station (systematic dataset: SYS), that was analysed first over the whole gauged period 1920-2018 (ALLSYS) and, secondly, subdivided on early (1920-1969; SYS1) and late (1970-2018; SYS2) datasets.

The documentary flood information is non-systematic data of censored type since only flows over a particular magnitude (commonly producing damages) are reported in documentary records. The minimum flood stage (perception threshold) are set according to flood magnitude spilling over urban areas disrupting human activities (communication, manufacture works) and/or producing damages (orchards, bridge, houses and ecclesiastic goods). The flooding of urban sensitive areas, or perception threshold, may change through time, according to the progressive human occupation of the riverine areas and the
socio-economic context (Benito et al., 2004). In the case of Zamora, the perception threshold did not register significant changes through historical time. The minimum discharge (1900 m$^3$/s) is required to flow overbank in the low urban neighbourhoods of Horta, Olivares and Frontis, and for cutting the main communication infrastructures next to the Duero River. The reconstructed historical flood discharges were added to the continuous gauged annual maximum flow records at the Porvenir and Carrascal stations.





The PeakFQ software uses the mentioned expected moments algorithm and a generalized version of the Grubbs-Beck test for
identifying multiple Potential Influential Low Flows (PILFs; Cohn et al., 2013). Low discharge annual values may have
excessive influence on the estimated frequency of large floods (Veilleux et al., 2014), and their identification of PILFS
improve estimated frequency of large floods. PeakFQ is well designed to treat both historical and systematic data but only
allows fitting a log-Pearson Type III distribution. A Gumbel distribution, commonly used in Spain, was fitted using the

AFINS software that applies a Maximum Likelihood Estimation method (MLE). This method has demonstrated a high
capacity to incorporate in the estimation process any non-systematic quantified data (Leese, 1973; Stedinger and Cohn,
1986). Visual matching of the plotting positions to the distribution curve and their statistical parameters were used to test the
goodness of fit. Confidence intervals of the fitted distribution indicate the range of discharges statistically possible based on
the available data.

**3.5 Analyses of the atmospheric circulation related to floods**

To provide a general context of the climate triggers of the larger floods exceeding the perception threshold of 1900 m$^3$/s, we
investigated the related atmospheric circulation based on the 20$^{th}$ Century Reanalysis climate data (Version 3/2c,
20CRV3/2c) from NOAA/CIRES. These dataset cover the period 1836 – 2015 at sub daily scale (3h) from around the globe
at 2°x 2° and provides relevant meteorological fields at different pressure levels (Compo et al., 2011). Thus, the climate

analysis is restricted to floods occurring during this period. First, we extracted several ensemble mean fields at daily scale
from the surface and troposphere level, such as geopotential height, wind components, divergence and specific moisture up
to 250 hPa. This information was used to carry out composite analyses of mean monthly anomalies (wrt: 1980-2010) to
describe the general situation during flood events. Then, we investigated the source of moisture triggering floods at daily
scale. To this end, we compute the vertically integrated water vapour transport (IVT) for the region (0-90°N; -100-20°E), as

suggested by Lavers et al. (2012) (Eq. 1), and plotting the wind vector.

$$IVT = \sqrt{(\frac{1}{g} \int_{1000\,hPa}^{300\,hPa} q\,u\,dp)^2 + (\frac{1}{g} \int_{1000\,hPa}^{300\,hPa} q\,v\,dp)^2} \qquad (1)$$

where $g$ is the gravitational constant (m/s$^2$), $q$ is the specific humidity (Kg/Kg), $u$ and $v$ are the zonal and meridional wind
component (m/s) and $p$ is the pressure (hPa). The IVT (Kg /sm$^{-1}$) is estimated to be between the sea level pressure and 300 h
Pa. To account for the rainfall-runoff transformation time in Duero Basin, we computed the averaged IVT over the 10

preceding days to the maximum peak discharge. Moreover, we used the *Katalog Der Grosswetterlagen Europas* (1881-
2004) to identify the predominant circulation pattern associated to each flood (Gestengabe and Werner, 2005). Since this
catalogue starts in 1881, we assigned a likely circulation type pattern for floods that took place before 1880 based on the
geopotential field from 20CRV3 (Table S2, Figure S3). Finally, we also used the NAO index to characterize the regional
influence of the North Atlantic atmospheric circulation variability (Hurrell, 1995; Brönnimann et al., 2008) on the historical

flood events. The NAO index reflects the difference in anomalies of the sea level pressure between Gibraltar (southwest





Iberian Peninsula) and Reykjavik (Iceland) stations, as has been used as a surrogate of temperature and precipitation winter pattern in the Iberian Peninsula (López-Moreno et al., 2011).

## 4 Results

### 4.1 Flood variability at decadal and multi-decadal time scales

Documentary flood descriptions of the Duero River in Zamora can be traced back to the 13[th] Century, although continuous records started since mid-16[th] Century. At that time, the main configuration of Zamora neighbourhoods, weirs, mills and bridges were similar to those of the late 19[th] Century and early 20[th] Century (Fig. 2a, b). This long-standing urban configuration allows a precise analysis of the sites and flooded areas as well as a qualitative reference of flood magnitudes over the last millennia. The morphological changes of river channel and banks are minor and mostly related to fluvial islands

and lateral bars stability by vegetation mainly over the last 30 years. For instance, the historical and present orthophotos shows stabilisation of a lateral bar next to the Cabañales mills, upstream of the Stone Bridge. The fluvial banks have remained at similar position according to the historical maps, at least over the last 300 years (Fig. 2a,b).

The documentary flood dataset comprises 69 flood entries over the period CE 1250-1871. These entries include mainly floods within catastrophic and extraordinary categories, as they were registered due to bank overflow and damages in

orchards, infrastructures and houses. The same analysis expanded towards the gauged period (since 1920) provides a flow discharge higher than 1900 $m^3$/s leading to similar inundation extent of reported overflows. The moving average analysis of temporal distribution using a 31-yr filter identifies flood-rich periods at 1600-1640, 1730-1750, 1770-1790, 1880-1910 (Fig. 3). Over the 20[th] Century, floods over 1900 $m^3$/s increased their frequency in the period 1935-1966. Later, some scattered large floods occurred in 1978-79 and 2001.

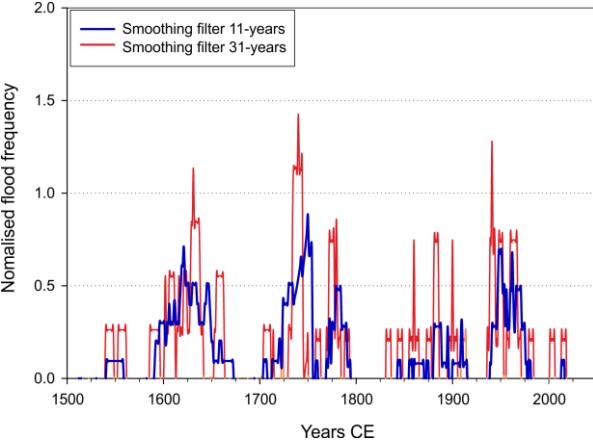


**Figure 3: Normalised flood frequency distribution of the documented number of floods (only catastrophic + extraordinary categories) of the Duero River in Zamora using moving average taking 11-yr and 31-yr data intervals. The anomalies for AD 1500–1880 were estimated from documentary records, and after 1980 includes gauged floods (staff gauge and continuous gauge) with discharge higher than 1900 $m^3$/s) gauged data.**





Large floods were produced mainly during the winter period (DEF; 66%), followed by spring (MAM; 28%), autumn (SON; 4%) and summer (JJA; 2%). This seasonal distribution was maintained in the different flood-rich periods established, with over 50% of floods concentrated in winter except for the period 1770-1790, where the highest number of floods occurred in spring. The highest concentration of severe winter events occurred during the 1630s, 1730s and 1960s with lack of reported large floods in the second half of the 17th Century and beginning of the 19th Century. The analysis of flood causes points to

persistent winter rain episodes, occasionally enhanced by snowmelt.

| Year | Date | Discharge (m$^3$/s) | Minimum Q (m$^3$/s) | Comments |
|---|---|---|---|---|
| 1258 | Dec, 12 | 3700 | 3540 | Level of postdoor in San Claudio |
| 1264 | - | 2700 | 2400 | Dueñas convent flooded |
| 1310 | Jan, 24 | | >2000 | Severe damage in old bridge |
| 1485 | Nov, 11 | | >2500 | Floods in Esla and Pisuerga |
| 1545 | Jan, 20 | | >2000 | Bridge arch destroyed |
| 1556 | - | | >2000 | Bridge arch and towers severely damaged |
| 1586 | - | 2800 | 2800 | Old St. Clara flooded. Archive destroyed |
| 1597 | Jan, 16 | 3200 | 3000 | 1.5 m depth at St. M. Horta. Larger than 1739-flood |
| 1611 | Feb, 28 | | >2000 | Arch and tower in bridge damaged |
| 1626 | Jan, 2 | | >2000 | Medieval bridge damaged |
| 1636 | Feb, 4 | 2500 | 2300 | 2.5 m long gap open at Cabañales wall |
| 1739 | March | 2700 | >2500 | Damaged 248 houses. Flood marks in Las Dueñas. Reported flooding in S. Claudio, S. Frontis, Santiago el Viejo |
| 1788 | Jan, 25 | 2550 | 2500 | Horta, Cabañales and Olivares zones flooded. 1.5 m depth at the Infantry Barracks |
| 1839 | Dec | | 1800 | Mark in Olivares |
| 1843 | Feb, 18 | 2500 | 2300 | Cabañales and Infantry Barracks in Horta flooded. Similar to 1788 flood |
| 1860 | Dec, 25 | 3450 | >3200 | Flood marks in S. Frontis, Dueñas, Pescado gate among others |
| 1872 | Jan, 30 | 1864* | | Villachica station. Marquina, 1941-44 |
| 1873 | Jan, 17 | 2200 | 1860* | Stage description at Iron bridge. *Porvenir |
| 1880 | Feb, 17 | 2370* | | El Porvenir staff gauge |
| 1881 | Jan, 14 | 2210* | | El Porvenir staff gauge |
| 1895 | Feb, 27 | 2380* | | El Porvenir staff gauge |
| 1900 | Feb, 13 | 2098* | | El Porvenir staff gauge |
| 1909 | Dec, 25 | 2155* | | El Porvenir staff gauge |
| 1911 | March, 11 | 1542* | | El Porvenir staff gauge |
| 1919 | Feb, 19 | 1620 | | El Porvenir staff gauge |

**Table 1. Reconstructed discharges of the major historical floods in Zamora. Historic discharges estimated at the scale of El Porvenir since 1860 are indicated by an asterisk (Marquina 1949b).**





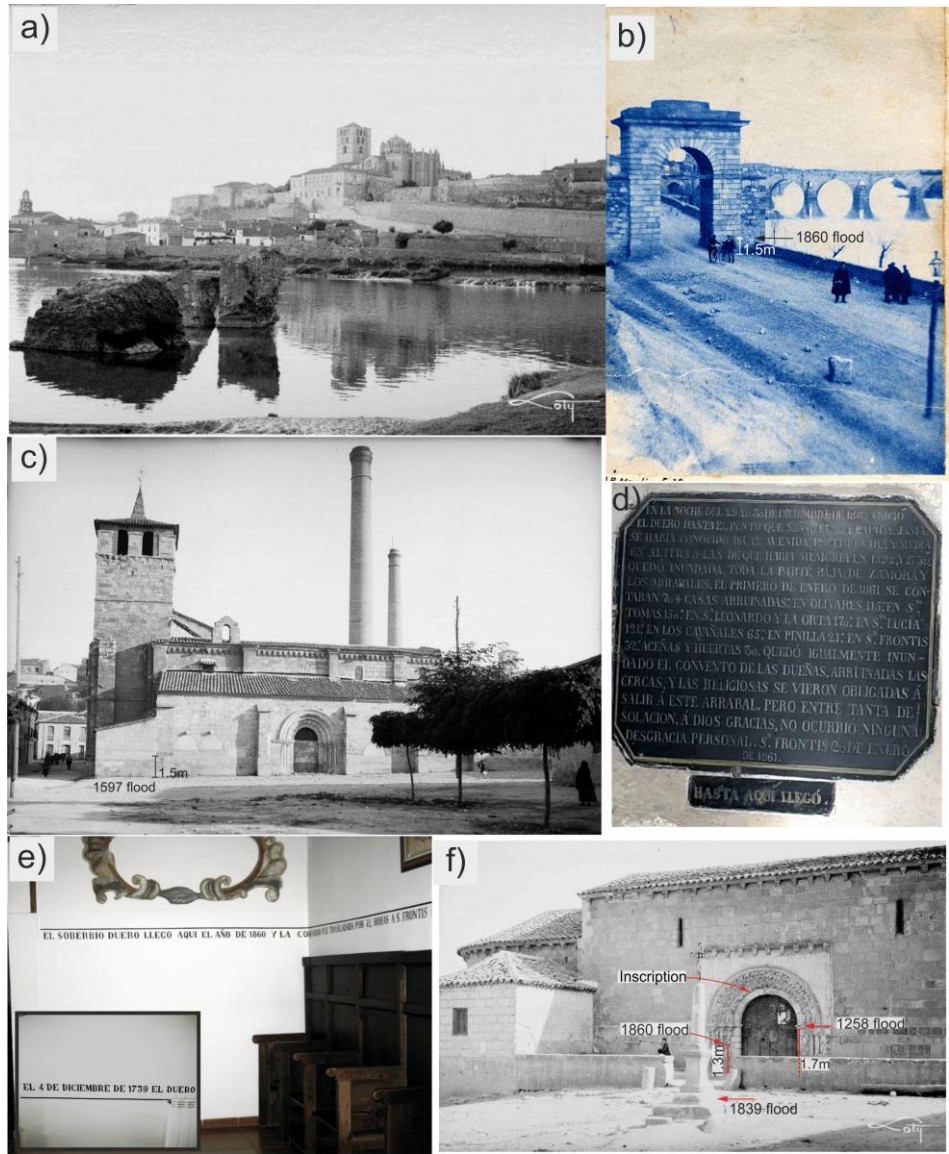


**Figure 4: Photos illustrating flood epigraphic marks and sites referred at written flood reports. a) View towards the Cathedral (center uphill) with remnants of the Old bridge destroyed by a flood in 1310. At the left opposite river margin are the Olivares mills and the tower of the San Claudio church (Photo António Passaporte 1927-1936; Source: Loty Archive-02471, Historic Heritage photo www.mcu.es/fototeca_patrimonio); b) View of the Pescado Gate where an epigraphic mark was at 1.5 m from the gate base (Source: Historical Provincial Archive). c) View of Santa Maria de la Horta reached by floods at least in 1597 and 1788. (Photo António Passaporte 1927-1936; Source: Loty Archive-02479, Historic Heritage photo www.mcu.es/fototeca_patrimonio); d) Epigraphic mark of the 1980 flood in San Frontis church. The description states that the 1860 flood was the largest of the last three centuries; e) Epigraphic flood mark of the 1860 flood in Las Dueñas convent at the refectory of the convent: "The superb Douro reached here in 1860 and the community was moved for 24 hours to San Frontis". Inset: Ephigraphic mark of the 1739 flood in las Dueñas convent at the same dining room but ca 1 m lower than the 1860 mark; f) View of San Claudio church between 1927-1936. The inscription in the arc refers to the "bad years of 1258" contemporaneous to the one at upper doorpost pointing to a flood (Marquina, 1949a,b). At the pedestal of the cross was an inscription to the 1839 flood but disappeared after restoration work (Photo António Passaporte; Source: Loty Archive-02464, Historic Heritage photo www.mcu.es/fototeca_patrimonio).**



## 4.2 Composite series of flood discharges

### 4.2.1 Historical flood peak levels and discharge determination

Over the last 500 years, the largest historical floods occurred in 1597, 1739, and 1860 exceeding 2800 m³/s. Previous large floods producing severe damage occurred in 1258, 1264, 1310, 1485 and 1586 with references to high flood levels and moderate damage. For instance, in San Claudio church an inscription at the arch (Fig. 4f) refers to times of "bad years" during the Kingdom of Alfonso X at the Hispanic Era 1297 (CE1259). Marquina (1941-1944) relates that inscription with a blurred mark located at the base of the arch, on the upper right doorpost (Fig. 4f), that is attributed to the Dec. 30, 1258 flood (628.7 m asl) matching a discharge of 3700 m³/s. During the 1310-flood the old bridge (early medieval) was destroyed although no references to landmarks indicating flood stage were found.

The 1860-flood is the largest at least over the last 500 years with evidence of flood stage on three epigraphic marks and five precise reports from sites at both sides of the Duero River (Fig. 4 b,d,e,f). The flood peak occurred at night of Dec 29 to 30 damaging 441 buildings in the city and another 263 in the suburbs. Fortunately, telegraphs received at the evening from cities upstream alerted on the flood severity and people living at risk areas were evacuated. In the village of Peleagonzalo (30 km upstream; Fig. 1c) 154 houses out of 160 were destroyed, and later rebuilt in 1862 on a nearby hill. Downstream of the Zamora's medieval bridge, the most reliable flood evidence are found in Dueñas Convent, San Claudio, Olivares watermill, Puerta del Pescado, and all pointing to a water stage of ca. 628.5 m asl (Fig. 5a; sites 1, 4, 3 and 9). At the upstream of the medieval bridge, flood elevation was 628.9 m asl at Santa Lucia (site 10) and at Iron Bridge (Zamora-Salamanca road; site 16) reaching San Leonardo and La Plata street (site 13). The epigraphic landmark in San Frontis (627.8 m asl; site 2 and Fig. 4d) is slightly below flood elevation reached in the Olivares landmarks at the opposite river bank. These 1860-flood high water marks fit a water surface elevation generated by the bi-dimensional model of 3450 m³/s (+/-100 m³/s).

Several documentary descriptions and epigraphic marks allow comparing the 1860 flood with previous flooding. For instance, in the Las Dueñas convent the 1860-flood epigraphic mark in the refectory is at 1.80 m above floor level whereas the 1739-flood is 0.75 m, meaning a water stage difference of 1.05 m (Fig. 4e). The San Frontis flood mark also refers to the 1860 flood as the largest compared to the previous 1597 and 1739 floods, although the description at the plate mentions a "1592-flood" which it seems to be a transcription error as noted by Marquina (1949a). The 1739-flood marks at Las Dueñas (627.45 m asl) and two minimum flood stages at Santiago El Viejo (626.6 m asl) and San Frontis-Cuesta de San Jerónimo (626.5 m asl) are associated with a discharge of 2700 m³/s. These flood stages are lower than 1597 flood reference reported at St. Maria de la Horta (628.35 m asl; Fig. 4c) and the minimum flood stages at Los Descalzos (628.2 m asl), San Juan de las Monjas (628.3 m asl). In the medieval bridge the spillway holes were coved by flood waters. The 1597-flood evidence matches a stage associated with a discharge of 3200 m³/s, meaning the second largest over the last 500 years (Fig. 6).



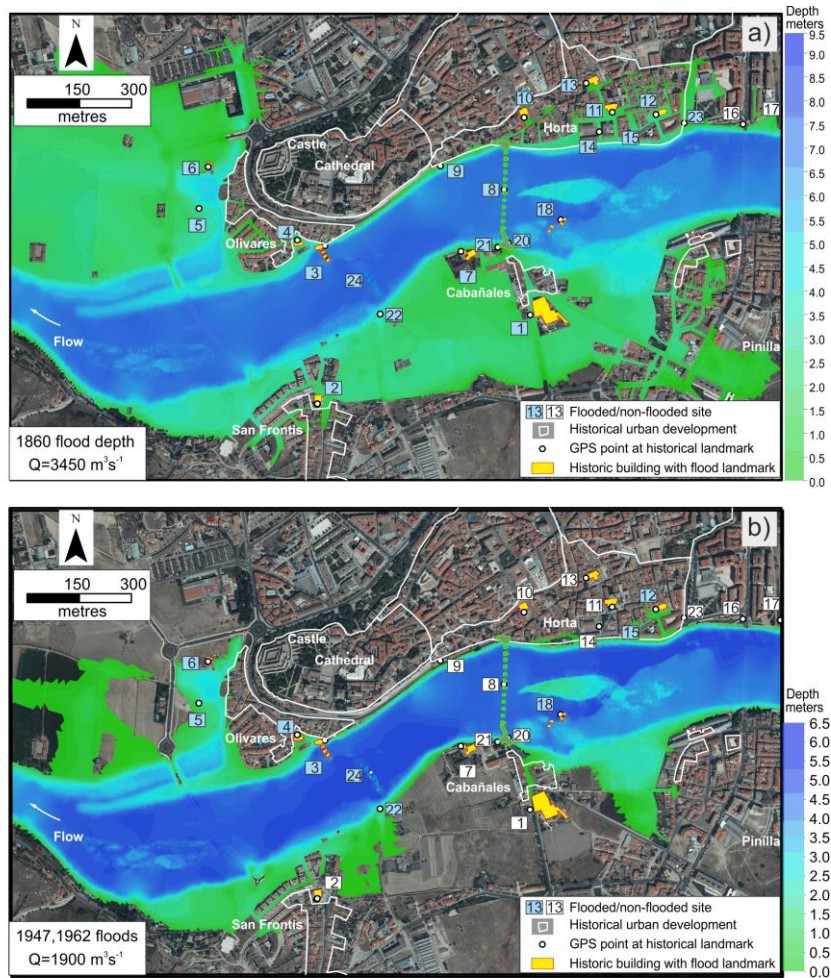

**Figure 5: Bi-dimensional hydraulic model results. a) Flood extension and depth for 3450 m³/s estimated for the 1860 flood and landmarks flooded (number in blue square). b) Flood extension and depth for a discharge of 1900 m³/s. Legend with names of landmarks in Fig. 2. Elevation (meters above sea level) reached by 1860-flood in Table S1. Aerial Orthophoto from the Spanish National Geographic Institute, IGN; https://www.ign.es.**




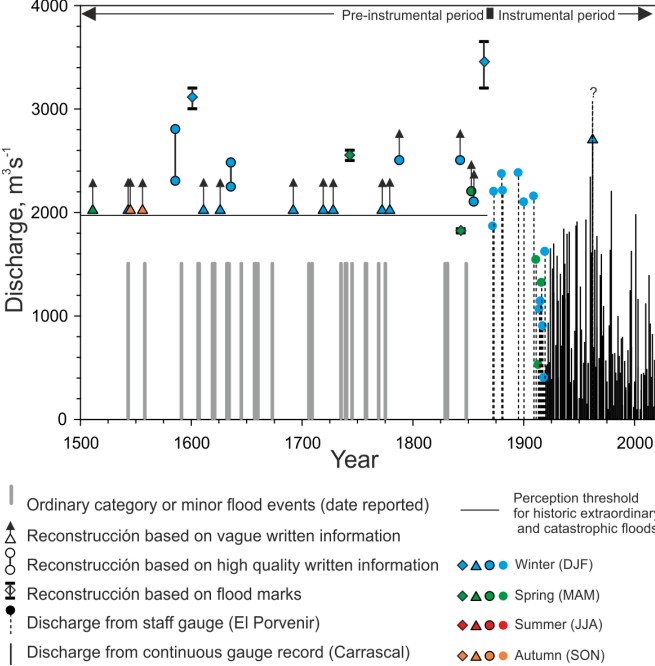

**Figure 6: The Duero River flood discharges 1500-2018 based on documentary (historic), water-level readings on staff gauge (El Porvenir) and continuous gauging (El Carrascal). Documentary floods with damaging overflows (catastrophic and extraordinary categories) exceeded a threshold discharge of 1900 cms. Discharge reconstruction were performed based on different description**
**details (from vague to high quality), flow depth at specific landmarks or sites and with epigraphic marks (types described in legend). Reported ordinary floods were estimated below discharge threshold. The flood season is indicated by colours as in legend.**

A second rank flooding corresponds to peak magnitudes exceeding 2200 m³/s that commonly produced overbank flow and damages and at the Horta and Olivares suburbs (Fig. 6). In this second magnitude rank, floods occurred in 1586, 1636, 1788, 1843, 1853, 1880, 1881, 1895, 1909 (2155 m³/s). The 1586 event flooded the old Santa Clara convent (Fig. 5, site 5) and
destroyed the archive that, assuming a minimum 0.5 - 1 meter water depth gives discharges between 2600-3000 m³/s. These flood magnitudes have caused damages in piers and towers of the Stone Bridge (e.g. 1636, 1880). The 1880-flood caused major damages in the bridge that it was reformed in early 20[th] Century reducing the number of arches from twenty-two to fifteen and enlarging the lightening arches or spillways (Rodríguez-Méndez et al., 2012). As a reference, these floods cover the bridge piers and spilled water through the lightening arches. In this magnitude rank, the flood cluster occurring during
the late 19[th] Century recorded at the Porvenir gauge (Fig. 6) is worthy of mention.

A third set of documented floods was reported to produce flooding at low city neighbourhoods and orchards surrounding convents and monasteries but without any description allowing any sort of discharge estimation. Modern flood analogues producing occasional inundation of low city areas (Cabañales and Olivares) and minor disruption of traffic activity are associated with discharges exceeding 1900 m³/s (Figs. 5b; Fig. 7).

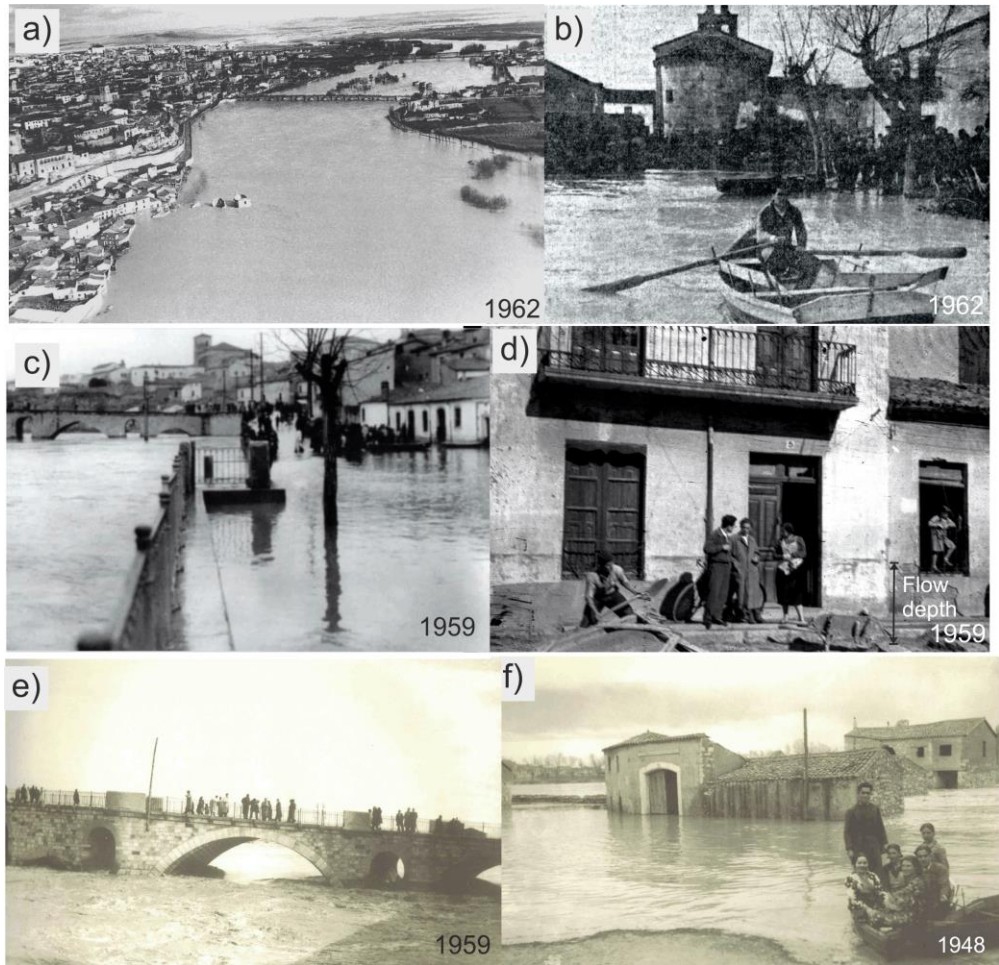


**Figure 7: The city of Zamora during flood episodes. a) Upstream view of the Duero river during the 1962 flood. At the left buildings submerged are the Olivares mill and the San Claudio church (Source: Archivo Gerardo Pastor Olmedo.T. III. La Historia Contemporánea.). b) A detail view of the 1962 flood at San Claudio (background) published by newspaper Imperio (5/02/1962) (available at https://prensahistorica.mcu.es); c) Horta Suburb (Mengue Avenue) and Stone bridge (background) during the 1959 flood (source: Memoria grafica de Zamora); d) Flood level reached by the 1959 flood at La Horta (source: Memoria grafica de Zamora); e) The Stone bridge during the 1959 flood with the water flowing through the bridge spillways (source: Memoria gráfica de Zamora); f) The Olivares suburb during the 1948 flood (source: Memoria grafica de Zamora). Note: the "Memoria gráfica de Zamora. Zamora: La Opinión-El Correo de Zamora, 2000. 396 p" published a collection of pictures from historical archives and individuals.**

**4.2.2 Modern flood records**

In the Carrascal station daily flows exceeding 2200 m³/s occurred in 1959-1960, 1961-1962, 1979, and 2001 (Qci 2140 m³/s) (Table S3). Intriguingly, the January 1962-flood recorded a daily discharge of 3071 m³/s that transformed to peak discharge results in 3200-3300 m³/s. Published photographs and descriptions in newspapers (Fig. 7a, b) show c. 1.5 m water depths at the Horta zone (Mengue Avenue), ca. 1.7 m next to the Cabañales mill, and the lower Cabañales neighbourhood (Table S1).

The documented flood stages agree with a discharge of c. 1900-2000 m³/s that suggest operative problems of the gauge



station during this flood. The seasonal hydrograph shows the January 1962 flood as the second peak of a sequence of five maxima that occurred from Dec. 1961 to Apr. 1962 (Fig. 8d). In the Villachica and Toro gauge stations (30 km upstream) this flood recorded daily discharges of 1729 (Jan 4, one day earlier) and 1531 m$^3$/s (Jan 5), respectively. A linear regression was fitted to daily discharges recorded in Carrascal and the Villachica stations over the period Dec. 1959-May 1960 (the

largest peak in Carrascal of 2343 m$^3$/s), and for the Carrascal and the Toro stations for the 1959-1960 and 1978-1979 periods. The calculated daily discharge for the 1962-flood in the Carrascal varied between 2100 m$^3$/s and 1940 m$^3$/s, which agrees with the estimated peak based on the photographic flow stage evidence.

Recent large floods are commonly produced during anomalous wet winters whose atmospheric conditions that may persist for weeks or even months producing hydrographs with multiple flow peaks (Fig. 8a, c). One of the most severe winters

occurred in 1935-36 with rains starting in late December extending unit April, giving rise to 12 peaks and a high flow stage over the whole season. Other flood season type (e.g. 1946-47) shows shorter frontal rain passage during the late winter and early spring (Fig. 8b). In this case, flow peaks are enhanced by rain on snow and snowmelt processes. In general, the largest flows peaks are produced by the passage of the second or third of those cold fronts, once the soils are saturated after previous rains.

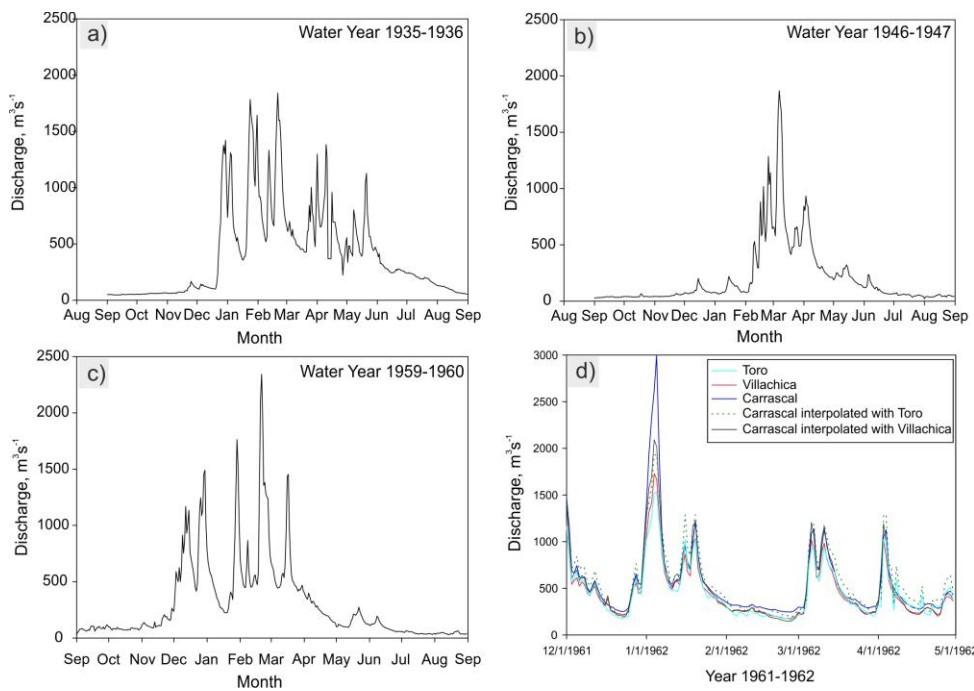


**Figure 8: Daily discharges showing multiple peaks during selected water years. a) 1935-36; b) 1946-47; c) 1959-60; c) Daily discharge recorded in the Villachica, Toro and Carrascal stations (see location in Fig. 1c. The recorded peak on Jan 5th 1962 in Carrascal was anomalously higher compared to Villachica and Toro records, and the peak was corrected using a linear regression.**





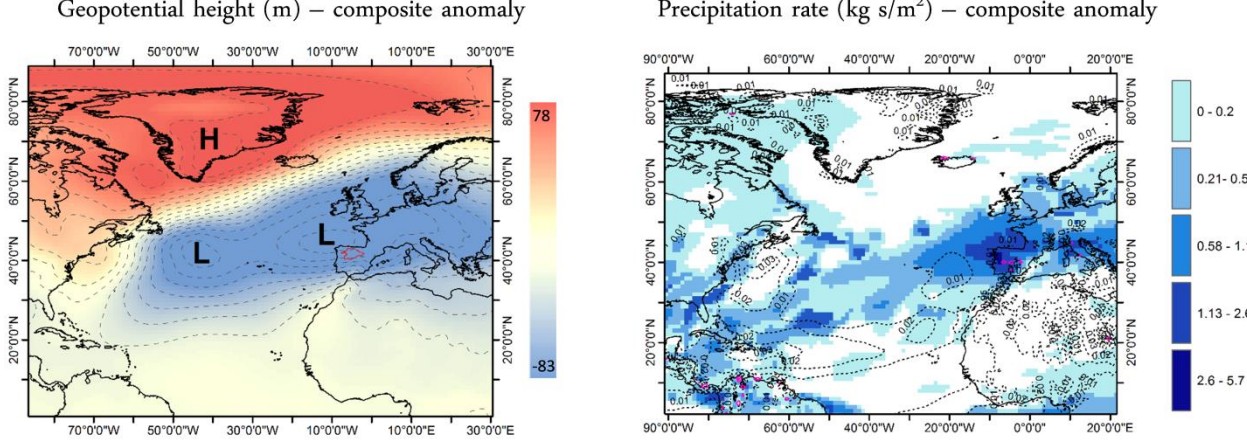


**Figure 9: Composite monthly anomaly of the geopotential at 500 hPa. b). Composite seasonal precipitation rate retrieved from 20CRV3 (blue-coloured) and composite divergence for the 10 prior days to each flood at 250 hPa (dotted), and convergence at sea level (purple line).**

### 4.3 Synoptic analysis and moisture transport

The meteorological predisposition and atmospheric circulation pattern analyses are focused on the major historical flood events since early 19[th] Century. The cyclonic west and south-west circulation are the main patterns related to intense floods accounting for the ~71% of the cases (Table S2). South-meridional type circulation seems to be less linked to floods (~17%), while through- (~5%) or high-similarity patterns (~5%) are marginal. Figure 9 shows the composite monthly geopotential at mid-levels (500 hPa) as well as the composite seasonal precipitation rate, wind vector and vorticity displaying negative / 475 positive values (surface level /250 hPa, respectively). Major flood events in the Duero basin are linked to intense cyclonic anomaly over the north-west of the Iberian Peninsula which extends to the mid-Atlantic Ocean (Fig. 9a). This pattern is well represented at surface level (Fig. S2), allowing the arrival of frontal system moving inland from easterly and warmer Atlantic positions. The result is the advection of relative warm moisture mass, favoured by stronger zonal winds that produce persistent rainfalls events. This situation is characteristic of negative NAO-like phases found during flood events, with a 480 mean 10-day composite NAO index of -0.9 ± 1.9. Looking at the specific moisture fields from the 10 days before the maxim peak discharge is recorded in Zamora city allows us to identify the source of the moisture responsible of each event. In combination with the wind fields, the responsible moisture is transported from low-to-mid altitude, along long and narrow bands from subtropical latitudes to the Duero Basin. The shape and the intensity of the integrated water vapour transported therefore suggest the existence of Atmospheric Rivers (AR, IVT > 250 Kg m$^{-1}$ s$^{-1}$; Wind velocity > 12.5 m/s). This has been 485 the case of 82% of the floods > 1900 m$^3$/s since the largest flood occurred in 1860. Thus, except for the flood which took place in 1962, AR-like structure has been detected over the Duero Basin (Fig. S3). In the Figure 10 is shown the example with the largest, the lowest (above 1900 m$^3$/s threshold) and more recent flood events, with maximum 10 days composite IVT > 550, 475 and 700 Kg m$^{-1}$ s$^{-1}$. Although landfall produced by the arrival of ARs is primordially related to orographic





conditions, rainfall seems to be enhanced by divergence in altitude (250 hPa) over Portugal and convergences at surface
level over the Duero Basin (Fig. 9b). Thus, according to the 20 CR records, the mean precipitation rate anomaly during flood
months in Duero Basin was +2.78 ± 1.3 (Fig. 9b, values presented in Table S2). This precipitation mostly occurred during
slightly warmer-than-normal months, as suggested by the mean temperature anomaly of +0.12 ± 1.0 (Fig. S2 and Table S2)
that it is consistent with the advection of moisture from tropical latitudes.

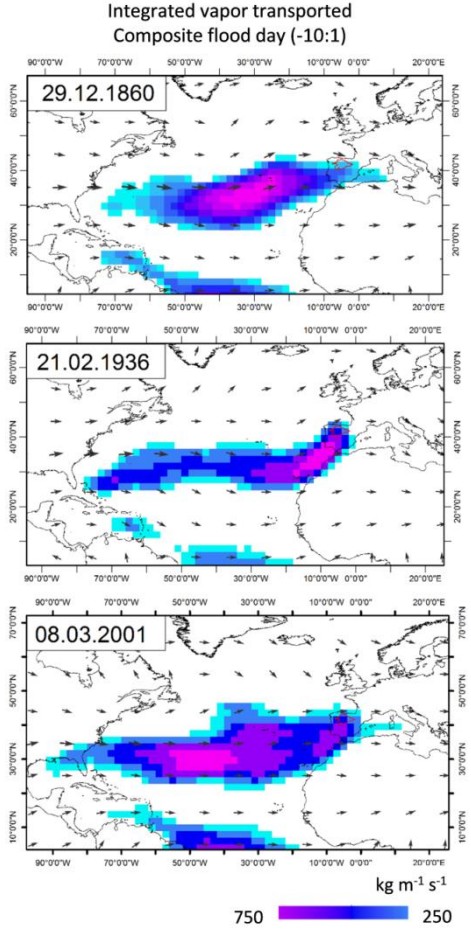

**Figure 10: Integrated water vapour transported averaged on the 10 days prior. Examples for historical flood took place in 1860,**
**1936 and recent flood in 2001.**

**4.4 Flood frequency analysis**

The stationarity tests (Lang et al., 1999 and 2004) of the combined documentary and instrumental (natural regime) flood
series with discharge equal or above 1900 m³/s show stationarity conditions over the period 1511-2018 (Fig. S2). The
number of floods decreased the frequency in the middle 18th Century overlapping the lower tolerance interval, but the overall
period is stationary for the perception threshold that includes extraordinary and catastrophic flood categories.





The FFA analysis was carried with two independent methods EMA and MLE combining non-systematic flood data (historic and pre-instrumental) and systematic continuous flood records (gauge station records). The Gumbel (two parameters) and Log-Pearson III distribution (three parameters) functions were applied to different datasets. Discharges calculated for

different return periods (T) are shown in Table 2. Figure 11 shows the plotting positions and the frequency curve fitted with LP3 distribution for different datasets showing good visual matching and within the confidence intervals. The confidence intervals in the upper tail of the distribution are narrower for the HISTO and ALLSYST datasets; moreover in the PRE-SYS analysis the 5% (lower) confidence interval is well constrained by the cluster of observed floods (Fig. 11a) plotted within 1-5% AEP (Fig. 11c). The subdivided systematic data sets (SYS1 and SYS2) show a poor performance at the upper tail of the

distribution with a wide range of discharges for the confidence intervals, particularly the upper 95%.

The HISTO data set fitted with LP3 distribution also provided the more realistic quantile values compared to other datasets. For instance, the largest flood during the 508 years record length, namely the 1860-event (3450 m³/s), is associated with a 500-yr recurrence interval (T) in the HISTO analysis, and with T >1000-yr and ~200-yr in the PRE-SYS and ALLSYS frequency curves.

The incorporation of historical and pre-instrumental data into the FFA results in a slight decrease of the magnitude of the flood quantiles, when compared with those calculated with the systematic record. In general terms, the systematic data provides realistic quantile values for return intervals lower than the 1% AEP, with unreasonable discharge values for the higher quantiles. The 2% AEP quantile (T= 50-yr) of the early systematic data set (1920-1969) gave a higher flood magnitude than the one from the latter systematic set (1970-2018) suggesting a decrease in the annual flood discharges

towards the late 20th Century.

The FFA analysis performed with the Gumbel distribution using the MLE method gives different discharge values to those obtained by the LP3 distribution, as expected. The differential performance can be evaluated in terms of discharge difference (DD in m³/s) obtained by the two distributions for a given quantile among different datasets. The most consistent results are obtained for the HISTO data set with differences of 20-75 m³/s for quantiles less than 0.2% AEP, reaching 155 m³/s in the

0.001% AEP flood. In the PRE-SYS dataset, the discharge calculated with Gumbel and LP3 distribution is similar for 1% and 0.5% AEP floods (T~70 years), but differences are wider towards frequent floods (e.g. 210 m³/s for the 25-year flood) and in the upper quantiles with a 300 m³/s in the 500-yr flood. In the ALLSYS dataset, the discharge range (DD) is wide over all quantiles and slightly increasing towards the 500-yr flood. The divergence on discharge calculated by the Gumbel and LP3 distributions is higher when applied to the subdivided systematic datasets within differences between 370-810 m³/s

in the 100-yr and 500-yr flood, typically used for flood hazard mapping. In summary, the FFA analysis using the historical dataset provided the most consistent results in discharges calculated for all flood quantiles using two different distribution functions (LP3 and Gumbel) and two independent fitting methods (EMA and MLE). The PRE-SYS data set showed good agreement on middle-term quantiles (T= 100-yr and 200-yr), whereas in the SYS data analysis results in 10-20% discharge differences that increased to 40-50% in the systematic sub-datasets (SYS1 and SYS2) for the higher quantiles (T= 500-yr).



**Figure 11: a)** Temporal representation of documentary and gauged floods. Grey shaded areas indicate censored flood records during the pre-instrumental period. The colour horizontal lines are perception thresholds over different time periods i.e., only floods exceeding the perception threshold discharge were recorded. The yellow diamond dots show discharges for documentary floods including a range of discharge uncertainty (vertical lines). The pink triangles are reported discharge from staff gauge observations at El Porvenir. The white dots are annual flows from the Carrascal gauge station. **b)** Log-Pearson 3 distribution fitted with historic flood events and gauged discharge (staff and continuous gauged). **c)** Idem fitted with staff and continuous gauge records; **d)** Idem with all continuous gauge records; **e)** Idem with gauged records over the period 1920-1969); **f)** Idem with gauged discharges over the period 1970-2018.





| Exceedance annual probability | Avg Ret. Inter T | Historic documentary, Staff Gauge & Systematic | | | Staff gauge (El Porvenir) and Systematic | | | Peak discharges in m³/s | | | | | | | | |
|---|---|---|---|---|---|---|---|---|---|---|---|---|---|---|---|---|
| | | | | | | | | All Systematic data only | | | Systematic data 1920-1969 | | | Systematic data 1970-2018 | | |
| in %/Value | (yrs) | LP 3 | Gumbel | DD years | LP 3 | Gumbel | DD years | LP 3 | Gumbel | DD years | LP 3 | Gumbel | DD years | LP 3 | Gumbel | DD years |
| 20/0.2 | 5 | 1240 | 1275 | 35 | 1450 | 1325 | 125 | 1390 | 1345 | 45 | 1760 | 1590 | 170 | 1080 | 1070 | 10 |
| 10/0.1 | 10 | 1650 | 1625 | 25 | 1875 | 1675 | 200 | 1810 | 1700 | 110 | 2160 | 1980 | 180 | 1445 | 1350 | 95 |
| 4/0.04 | 25 | 2140 | 2070 | 70 | 2330 | 2120 | 210 | 2335 | 2140 | 195 | 2520 | 2475 | 45 | 1960 | 1700 | 260 |
| 2/0.02 | 50 | 2475 | 2400 | 75 | 2610 | 2450 | 160 | 2720 | 2470 | 250 | 2700 | 2840 | 140 | 2380 | 1970 | 410 |
| 1/0.01 | 100 | 2790 | 2720 | 70 | 2840 | 2775 | 65 | 3090 | 2800 | 290 | 2830 | 3200 | 370 | 2825 | 2230 | 595 |
| 0.5/0.005 | 200 | 3070 | 3050 | 20 | 3030 | 3100 | 70 | 3450 | 3125 | 325 | 2920 | 3575 | 655 | 3300 | 2490 | 810 |
| 0.2/0.002 | 500 | 3410 | 3475 | 65 | 3230 | 3530 | 300 | 3910 | 3550 | 360 | 3000 | 4045 | 1045 | 4000 | 2830 | 1170 |
| 0.1/0.001 | 1000 | 3640 | 3795 | 155 | 3360 | 3855 | 495 | 4250 | 3880 | 370 | 3050 | 4405 | 1355 | 4550 | 3090 | 1460 |

**Table 2: Flood quantiles for different exceedance annual probabilities of the Duero River in Zamora calculated for different**
**datasets applying the Expected Moments algorithm to fit a Log Pearson type 3 distribution and the Maximum Likelihood Estimator Methods to fit a Gumbel distribution. DD represent the difference in years of the calculated discharge obtained for each quantile by the fore mentioned distributions. T: Average Recurrence Interval (in years).**

## 5 Discussion

### 5.1 Multi-decadal flood patterns and climate variability

Documentary flood records in Zamora are scarce during the early medieval period coincident with the late Climatic Medieval Anomaly that is characterised by warm temperatures and high hydrological variability. The limited flood data suggest, however, the occurrence of exceptional floods such as the 1258 and 1264 events, the former with magnitude similar or exceeding the 1860 flood as it is interpreted from inscription at the doorpost in St Claudio church. The 1258 flood was exceptional in terms of peak flow, but also in extension affecting other Iberian Atlantic basins, at least the Tagus and
Guadalquivir Rivers (Benito et al., 2003). The lack of information of catastrophic flood between 1270 and 1500 is common to other Iberian rivers, although it is likely related to the discontinuity of written reports and preservation of documentary archives prior to 14th Century (Barriendos and Rodrigo, 2006).

The temporal distribution of floods over the past 500 years shows at least six flood-rich periods of 20-40 year duration, commonly separated by ~60-year periods with scarce numbers of large floods. Climatically, this period corresponds to the
Little Ice Age (1500-1850) that in Iberia is characterised by cold conditions with alternating wet-dry phases (Oliva et al., 2018). Flood-rich periods were identified at 1600-1640, 1730-1750, 1770-1790, 1850-1880, 1924-1948 and 1960-1980 (Fig. 3) that coincides with flood episodes in Atlantic Iberian Rivers (Benito et al., 1996; Benito et al., 2003; Barriendos and Rodrigo, 2006) and overlap in time with flood periods described in western European regions (Blöschl et al., 2020). The temporal pattern of these flood clusters suggests a multidecadal natural variability on the atmospheric circulation affecting
both flood frequency and magnitude (Nobre et al., 2017). In western Iberia, excess of winter precipitation is related to a





southern position of Atlantic storm tracks occurring during the negative mode of North Atlantic Oscillation (NAO) (Trigo et al., 2014).

A detail hydroclimatic analysis driving these flood-rich periods reveals complexities and dissimilarities among them. The first two periods (1600-1640, 1730-1750) were also identified as decades with frequent dry years in spring (needed for
agriculture) according to reported prayers and novenas in churches and processions for rain usually undertaken in March and early April (Álvarez-Vázquez, 1986). The second two periods (1770-1790, 1850-1880) were dominated by overall wet winter years, albeit that the total of rainy years were never more than a quarter of the drought years (Álvarez-Vázquez, 1986). In the Douro River in Porto these two periods were also identified by their anomalous frequency and severity of floods (Alcoforado et al., 2021; Amorim et al., 2017) that includes the largest flood on record (Dec 4-6, 1739) that reached a
stage of 12 m at Dom Luiz I Bridge (Loureiro, 1904; Taborda, 2006).

In western Iberia, the relationship between temperature and precipitation tends to be negative, as cyclonic conditions are related to moist and relatively warm air masses from the Atlantic. Indeed, there is a total absence of large floods during the Maunder Minimum period (1645-1715), a period with lower air temperatures, that intriguing reveal a link between low solar activity and decreased flood frequency in the Iberian Peninsula (Vaquero, 2004). In contrast, periods with more frequent
floods in the western Iberian region coincide with transitions to cool and wet conditions associated with a southward migration of westerlies (Benito et al., 2015b).

In terms of flood magnitude, the Duero data suggest that the largest floods occurred at the onset and final stages of the Little Ice Age. In both instances, these floods are related to persistent rainfalls occurring during winter months. Interestingly, during the 16[th] and early 17[th] Centuries catastrophic floods occurred mainly in January and extraordinary floods in early
spring (March), whereas at the final stage of the LIE the largest floods trend to occur between February and March (with the exception of the 1860 flood).

The flood magnitudes over the 20[th] Century may be biased by the environmental changes (agricultural transformation in the early part of the century) and reservoir construction (since 1950s). The 1924-1948 flood-rich period took place during the Early Twentieth Century Warming (ETCW; 1910s-1940s) a period of strong internal variability of the climate system, which
featured an anomalous warming of the Arctic region impacting climate in North Europe (Brönnimann, 2009). Two periods (1924-1927; 1935-1941) of extraordinary winter floods ($Q > 1250$ m$^3$/s) took place during the ETCW could arguably make this event relevant also for future analogues warming phases (Ballesteros-Cánovas et al., 2019). Over the late 20[th] Century, the frequency of extraordinary flooding has decreased and mostly occurred in late winter-early spring, that confirms the delay of flood peaks under a warming climate in Europe (Blöschl et al., 2019).

**5.2 Atmospheric-ocean interaction leading to catastrophic flooding**

During the 18[th] Century the largest flood occurred on Dec 5, 1739, with an estimated peak of 2700 m$^3$/s. In this winter, a previous peak flow was already reported on Nov 1, with a likely discharge of 2000 m$^3$/s, and at least another peak documented in April 1740 with lower magnitude (~1800 m$^3$/s). The reconstructed monthly NAO index (Luterbacher et al.,





1999) shows negative values on the month previous to peaks, namely -1.27 (Oct), -1.47 (Nov) and -0.38 and-0.55 (March
and April) that, together with -2.38 and -1.14 during Jan and Feb, shows persistent meridional atmospheric zonal flow during
that exceptional winter. Other winter with several peaks was 1931 recorded in Jan and Mar (-2.08 and -1.16 NAO). Also
1881 floods in Jan 14 (-3.6) and May (-1.42). The Dec 29, 1860 flood was preceded by negative NAO index in Nov (-3.44)
and Dec (-2.14) and continued into Jan 1861 (-0.56). In the Tagus River, similar large floods have been associated with a
very high frequency of negative NAO mode during the initial 20-25 days before the flood peak (Salgueiro et al., 2013).

The negative NAO-like phases allow the arrival to Iberian Peninsula of frontal system bringing warmer and enriched
moisture air masses. Our analyses highlighted that the source of this moisture is the Caribbean Sea, and that is the so-called
Atmospheric River structures (Ralph et al., 2017; Dacre et al., 2014; Waliser and Guan, 2017). Thus, most of the major
floods recorded in Zamora (88 %) were linked to the occurrence of these phenomena. A similar mechanism for moisture
input has been associated to intense floods in Portugal (Trigo et al., 2014) and to others large European rivers draining to the

Atlantic Ocean (Lavers and Villarini, 2015; Ballesteros-Cánovas et al., 2019). Rainfall events linked to the arrival of the ARs
are generally related to the uplift forced by orography (Ralph et al., 2006), which is consistent with the mountain reliefs
surrounding the Duero Basin. Although the frequency, position, and magnitude of ARs depends on planetary-scale
phenomena (Ralph et al., 2011), the moisture transport capacity may be enhanced under climate change condition as
consequence of an increase in the water-holding capacity of the atmosphere (Lavers et al., 2013), which could have

consequences of climate predisposition for floods in the Duero Basin.

Our analyses also pointed that floods were linked to warmer-than-normal air temperatures, as identified in the composite
analyses in Duero basin (Figure S2). This is consistent with the cyclonic circulation and the advection of template air mass,
especially with a south-west orientation (Trigo et al., 2014). Thus, our results suggest that at least 70 % of the major floods
recorded were related either to west or south cyclonic circulation patterns and at least in 14 % of cases were related to

south/west meridional circulation patterns. This implies that snowmelt and / or rain on snow from the surrounding mountains
could have contributed for additional runoff (Stewart, 2009). Therefore, these mechanisms were related to main triggering
mechanisms of torrential floods in mountain streams in Duero Basin (Ballesteros-Cánovas et al., 2015; Morán-Tejeda et al.,
2019). Moreover, during wet winters, characteristic of negative NAO-like phases (López-Moreno et al., 2007), a high
moisture soil content prevalence in large portion of the basin, could have enabled a recharge of the groundwater system and

therefore favoured the direct rainfall-runoff transformation (Berghuijs et al., 2019; Benito et al., 2010; Benito et al., 2011).

**5.3 The significance of past floods in flood hazard analysis**

The historic city of Zamora is highly sensitive to flood hazards and weather extremes as both have direct impacts on
architectural, patrimonial assets, and cultural landscapes. Reported evidence of flood incidence provide a rich event
catalogue with description of more than 88 historical floods over a period of 760 years, as well as, 17 pre-instrumental
observations on a gauged water-level scale and 99 year of gauged continuous records.



The basic hypothesis in flood frequency analysis (FFA) incorporating historical information (non-systematic) is that a certain perception threshold of water level exists and all the exceedances of this level over a specific time interval have been recorded (Benito et al., 2015a). Perception thresholds are typically related to river morphological elements (e.g. river banks, dikes) above which overflow produces damages reported in municipal and ecclesiastic act books (Stedinger and Cohn, 1986;

Frances et al., 1994). For instance, in the Duero's documented floods, the perception threshold is the flood discharge overflowing riverbanks at Olivares, Horta, Frontis and Cabañales neighbourhoods that is ~1900 m$^3$/s. Each year, therefore, the Duero River was characterised as having a peak discharge either exceeding, or not exceeding, that perception threshold. In the qualitative historical classification without any further damage descriptions such events correspond to extraordinary floods (Barriendos and Martín-Vide, 1998). Catastrophic floods involve higher damages that are typically recorded in

epigraphic marks and/or reports of impacts on historic buildings. The scarce urban development of Zamora until the second half of the 19$^{th}$ Century allows a temporal stability (from 1511 to 1872) of this perception threshold, that it was confirmed for modern flood analogues of historical extraordinary and even catastrophic floods. Over the 408 years prior to the continuous gauged record (1511-1919) there is evidence of thirty floods exceeding the perception threshold (1900 m$^3$/s), whereas eight floods exceeded that discharge over the gauged period (1920-2018).

More important are the consequences of possible climate-related non-stationarity for estimating flood quantiles (Milly et al., 2008). The alternation of flood-rich and flood-poor periods identified during the last 500 years implies differences in the statistic values over time. However, the problem affects both historic and the modern gauge records with the advantage that the former includes a longer-term flood dataset (rich and poor flood periods) whereas there may be a bias in gauged data for specific flood patterns. For instance, seven catastrophic floods were recorded over the period 1920-1969 whereas only two

occurred over 1970-2018, that results in strong differences in the 25-yr flood with 2500 m$^3$/s and 1650 m$^3$/s respectively. The gauged registers indicate a trend to decrease the magnitude of extreme floods since 1970s; however, flood events such as 2001 (2075 m$^3$/s) and 2013 (1654 m$^3$/s) illustrate the occurrence of extreme flooding despite the peak discharge attenuation by reservoirs. The overall gauge record (1920-2018) estimate the 25-yr flood in an intermediate value of 2105-2300 m$^3$/s. In the case of long-term historical records, multiple flood-rich and flood-poor periods are combined giving averaged estimates

for the flood quantile discharges (e.g. T 25-yr: 2140-2070 m$^3$/s).

Due to the extensive historical records of the area, flood hazard assessment can be performed integrating different flood sources, datasets and time scales. Moreover, the implementation of two independent methods (EMA and MLE) for fitting regression models to censored data together with two distribution functions (LP3 and Gumbel) allows testing the robustness of results for low probability quantiles. In the case of the 1% AEP flood (T=100-yrs), both the HISTO dataset (511 years)

and the PRE-SYST (147 years) result in a similar discharge (2800 m$^3$/s) whereas the SYS dataset (99 years) provides a range of 2800-3100 m$^3$/s. In the upper tail of the distribution, the 0.2% AEP flood (T= 500 years) based on HISTO data is well constrained to 3410-3475 m$^3$/s, whereas a wide discharge range is obtained with PRE-SYST (3230-3530 m$^3$/s) and it is even wider with SYST dataset (3550-3910 m$^3$/s). The historical flood record indicates that, at least over the last 511 years, only one flood reach a discharge of 3450 m$^3$/s (1860 flood), which is within the range of the 0.2% AEP flood, whereas the SYST


dataset overestimate the quantile discharge (3910 m$^3$/s). In even rarer floods, the 0.1% AEP based on the HISTO dataset was estimated in the range of 3640-3795 m$^3$/s which fits the 3700 m$^3$/s estimated for the 1258-flood, albeit that historical flood was not included in the frequency analysis.

Previous studies of FFA under non-stationary framework have been applied to documentary flood datasets using climatic (e.g. NAO) and environmental indices (reservoir index) as external covariates (Machado et al., 2015). In such non-stationary

approach, historical flood frequency analysis over the last 300 years shows fluctuations on the estimated flood quantiles at decadal scale responding to a combination of multi-decadal cold-warm cycles and interannual ocean-atmospheric interactions. However, such analysis is not easy to implement (e.g. (López and Francés, 2013) and it is beyond the aim of the present study. Alternative options using climate models and scenarios may generate even higher uncertainties on flood hazard and risk planning studies (Serinaldi and Kilsby, 2015). Lins and Cohn (2011) suggested that it is preferable to

continue with simple (stationary) statistical models with well-understood limitations than use sophisticate models whose correspondence to reality is uncertain. In this debate, historical hydrology contributes, providing robust data of real floods that occurred at centennial scale, information about timing and persistence of flood clusters and identifying the relationships of the largest floods with low-frequency atmospheric circulation and other environmental drivers. In summary, a long-term FFA framework under stationary models provides good average values supported on flood extremes beyond century-scale

climate cycles.

## 5.4 Public perception and risk culture

Flooding of the Duero River has been a recurrent problem for the city of Zamora with most reported incidents related to the medieval bridge, watermill facilities, ecclesiastic buildings, farms and houses. Despite the abundant flood documentation, details on the relative importance of each historic flood are not always available, and only for the most catastrophic events

are there precise references to flood levels on epigraphic marks. The best-preserved epigraphic marks are located inside San Frontis Church and Dueñas Convent, which became a cloistered convent with difficult access for the public. Epigraphic marks are important elements for public perception in central Europe (Brázdil et al., 2005; Herget and Meurs, 2010; Wetter et al., 2011) and France (Cœur and Lang, 2008) which are maintained as part of the historical asset. In Zamora flood marks are not well known to the public and, in the modern urban expansion, some flood marks were removed during restoration

works of architectural assets. For instance, the 1860-flood mark disappeared from the old city gate (Puerta del Pescado) when, due to traffic problems, the old monument was moved to a new location in 1909. Curiously, in a Municipal council meeting (Jan 29, 1908), a city councillor (Mr. Calonge) proposed to preserve the 1860-flood mark at its place for the public memory of that extreme flood. In that time five floods over 2000 m$^3$/s were recorded in a thirty-year period (1880 to 1909) that kept alive the public perception of flood risk and their socio-economic consequences. It also corroborates a well-

established observation that occurrence of a damaging flood improves awareness, social learning and enhances adaptation (Di Baldassarre et al., 2015). Conversely, over the last 40, only the 2001 flood exceeded 2000 m$^3$/s, and since the 1980s there is strong socio-economic pressure to expand urban occupation in the San Frontis, Olivares and Cabañales suburbs.





Fortunately, the remarkable work on historical flood data collection done by Marquina (1949a,b) preserved the details on the location and position of epigraphic marks no longer at their original sites.

Within the framework of the European Flood Directive (2007/60/EC) EU members states are requested to map flood hazards (typically 100-yr and 500-yr floods) as well as consider the effects of climate change on flood risks. The climate effect on flood hazards is complex and not ubiquitous within a world that increases population, exposure and vulnerability (Kundzewicz et al., 2019). Recent developments on climate change science and adaptation actions focus on win-win strategies for sustainable climate action. The study of past flooding, either historic or palaeofloods, can be a direct guide to

flood possibilities for adaptation actions to extreme flooding under climate change, as it reveals what is actually possible under conditions of centennial climate variability. The flood magnitudes neither result from complex climate models nor from probabilistic language, difficult to understand by the general public. Common sense holds that what has really happened can happen again (Baker, 2008). This approach based on past flood occurrences also serves to increase public confidence in any proposed solution that ultimately involves a large economic or social expense for hazard mitigation.

**6 Conclusions**

Climate change effect on local flood magnitudes is becoming a major problem in flood hazard mapping due to model uncertainty and multiple possible scenarios. There is a need for better identification and guidance for tools for a robust adaptation to future flood risk. In this work, we demonstrate that the reconstruction of flood histories under climate variability beyond the temporal length of observations provide realistic insight into future floods under worst case scenarios.

In Zamora (Spain), a long-term flood record of the Duero River was collated from documentary archives (1256-1871), early water-level observations (1872-1919) and gauged data (1920-2018). Early flood records were discontinuously reported but since c.1500, flood events are well documented. Documentary flood information includes narrative descriptions (annals, chronicles, and memory books), epigraphic marks, newspapers and technical reports. The urban configuration at the riverside area has been stable since the 14th Century which implies continuous, homogeneous and comparable data over the

last 500 years. Our key findings are:

- There is documentary evidence of 69 flood registers during the period 1511-1871, with 15 catastrophic floods, 16 extraordinary floods and 38 ordinary floods. Catastrophic and extraordinary floods typically produced overflow of urban areas, typically exceeding a discharge of 1900 m$^3$/s, based on a two-dimensional hydraulic model calibrated with gauge records.

- The largest floods over the last 500 years occurred in 1860 (3450 m$^3$/s), 1597 (3200 m$^3$/s), and 1739 (2700 m$^3$/s). Historic flood magnitudes were greater than the ones recorded in the early water readings (largest in 1895 with 2380 m$^3$/s) and in the gauging station (largest in 1960 with 2360 m$^3$/s).

- Over the historical time, flood-rich periods were identified at 1600-1640, 1730-1750, 1770-1790, 1880-1910, the first two in decades of frequent drought, and the latter in overall wet winter years. In the 20$^{th}$ Century flood frequency


increased at 1935-1960 overlapping the Early Twentieth Century Warm Period. Later, the flooding occurred punctuated on time, namely in 1978-79 and 2001.

- Floods in Duero Basin mostly occur during negative-like NAO phases that force frontal system to travel in meridional latitudes, bringing moisture and warmer air masses from the tropical latitudes in the so-called Atmospheric River structures.

- We demonstrate that the temporal extent of the flood dataset is a critical factor in the quality of the flood frequency results. The most consistent results, independently of fitting methods and distribution functions, corresponds to the historical dataset (also includes pre-instrumental and systematic records) showing calculated discharges within a narrow range even for high quantiles (1000-yr flood). Combined pre-instrumental (water-level readings) and systematic gauge data provided steady results for robust for 100-yr flood with increasing uncertainty in the 500-yr flood. In the systematic

dataset results were uncertain in the 100-yr and 500-yr floods, despite the almost 100 year-long datasets.

- Since 1970s the frequency of extraordinary floods (>1900 m$^3$/s) declined, although floods above historical perception threshold occurred in 2001 and 2013. The decreasing frequency of extraordinary floods, relatively common at the end of the 19$^{th}$ Century and first half of the 20$^{th}$ Century, may be responsible for a lower flood risk perception. The fact that some extraordinary floods occurred at the beginning of the 21st Century, and historical analogues of extreme flooding

during drought dominated periods, demands a higher degree of flood education and spatial planning according to consolidate flood hazard analysis.

- Although the likelihood of future floods is uncertain with conventional downscaling climate models, the extending of flood records beyond cycles of past climate variability provide further understanding on possible flood sizes, and calculated magnitudes of flood quantiles to guide on low-regret adaptation decisions, and to improve public perception

of extreme flooding.

**Code and data availability**

The data used in this article can be obtained by contacting the corresponding author.

**Supplement**

Supplementary data to this article can be found online at: https://doi.org/...........

**Author contributions**

GB and MM designed the research and applied for funding acquisition. OC carried out the two-dimensional hydraulic modelling. JB-C performed the analysis of the atmospheric circulation in relation to historical floods and its



hydrometeorological interpretation. M.B. and MM collected and analysed the documentary flood database. MM collected and analysed the urban history and its relation to flood perception through time. All authors interpreted results. GB, JB-C 760 and M.M. wrote the first draft and all authors contributing reviewing and editing the paper.

**Competing interests**

The authors declare that they have no conflict of interest.

**Acknowledgements**

We acknowledge the hydrological data and documentary support provided by Jose Angel Martinez Pérez, Head of Operation 765 and Management Service of Iberdrola (Hydropower Company) in Carbajosa de la Sagrada (Salamanca), and to Yolanda Diego Martín, Director of Documentation Management of the Iberdrola Historical Archive in Ricobayo (Zamora). Mikel Calle Navarro provided field assistance for GPS survey of historical landmarks and GPS point processing.

**Financial support**

This project was supported by the Fundación Biodiversidad (MITERD) through the project "Regional Information on 770 Climate Change and floods for the design adaption and safety analysis of sensible infrastructures" and the project "Assessment and modelling eco-hydrological and sedimentary responses in Mediterranean catchments for climate and environmental change adaptation" EPHIMED (CGL2017-86839-C3-1-R) funded by the Ministry of Science, Innovation and Universities co-financed with FEDER funds.

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
