# Peer review of "Enhanced flood hazard assessment beyond decadal climate cycles based on centennial historical data (Duero Basin, Spain)"

_Hydrology and Earth System Sciences, 2021_

## Referee Comment (RC1)

**Enhanced flood hazard assessment beyond decadal climate cycles based on centennial historical data**

Porto, 24 august 2021, by Inês Amorim

The article is an excellent methodological approach to an increasingly important theme: the study of past flood occurrences and how it serves "to increase public confidence in any proposed solution that ultimately involves a large economic or social expense for hazard mitigation" (lines 708-709). I consider that the main contribution of this article is the systematic use of proxy and instrumental long-term data, crossing the two dimensional of hydraulic modelling (referred in 3.3). I considered this article a pilot project of a methodology application that could be replicate in other cases.

A first suggestion to clarify the extent of this article is the inclusion, in the title, the mention to the space: the river and place (River Douro and Zamora). Indeed, the article is mainly an approach to the analysis of the river Douro floods in the Spanish area, between Zamora and the first dams, that were constructed in 1960s after Zamora town. The article is a case study, a methodological research about a particular section of Douro River, with its own characteristics (as was explain in the contain).

A second suggestion is the possibility to add extreme dates in the title, even if is difficult to fixed the scope, because sometimes are mentioned (and effectively was study) the last 500 years, and in other occasions the period between 1250-1871, maybe because they faced «a non-continuous dataset between 1250 and 1545» (I understand the expression «centennial» in the title).

The authors made a remarkable comparative approach putting its case in a larger frame. I suggest another comparative analysis using an article that tried to estimate the frequency of extraordinary floods of Douro River in the Portuguese territory, till its estuary. (Silva, J.D. da; Oliveira, Manuel de Sousa - *As cheias na parte portuguesa da bacia hidrográfica do rio Douro, ps/p/*. Available https://grupo.us.es/ciberico/.../porto2diasdasilva.pdf.). It could be important to insist in a comparison between rainfall contrast characteristics consequences (line 181 and further), i.e., the flood peaks contrast between Régua and Porto at the lower basin of Douro, with Valladolid and Zamora and its consequences. The period 1250-1871, in which were identified 69 floods (including ordinary ones), is a number very low if compared with the floods of the Douro in Porto just for the period between 1727 and 1799, in which were found 54 floods (see the quoted article by the authors). Perhaps this increasing number of floods was related to the tidal peaks and the siltation of the estuary, but this contrast of occurrences could open an outlook about long Douro River course behavior, before and after dams' construction.

A final remark: line 575, authors wrote "includes the largest flood on record (Dec 4-6 1739) that reached a stage of 12 m at Dom Luiz I Bridge (Loureiro, 1904; Taborda, 2006)". The bridge doesn't exist in 1739 (only constructed between 1881-1886) but Loureiro (and Taborda quoting Loureiro), used the currently existing bridge as a mark (rebuild the sentence will be enough).

---

## Author Response (AR1)

**Reviewer comments Inês Amorim**

REVIEWER COMMENT:

The article is an excellent methodological approach to an increasingly important theme: the study of past flood occurrences and how it serves "to increase public confidence in any proposed solution that ultimately involves a large economic or social expense for hazard mitigation" (lines 708-709). I consider that the main contribution of this article is the systematic use of proxy and instrumental long-term data, crossing the two dimensional of hydraulic modelling (referred in 3.3). I considered this article a pilot project of a methodology application that could be replicate in other cases.

RESPONSE:

We appreciate very much the words of the Prof. Amorim as a recognised expert on the study of historical flood events and social perception of flood risks.

REVIEWER COMMENT:

A first suggestion to clarify the extent of this article is the inclusion, in the title, the mention to the space: the river and place (River Douro and Zamora, Spain). Indeed, the article is mainly an approach to the analysis of the river Douro floods in the Spanish area, between Zamora and the first dams, that were constructed in 1960s after Zamora town. The article is a case study, a methodological research about a particular section of Douro River, with its own characteristics (as was explain in the contain).

RESPONSE:

Thanks for this comment. We initially omitted the river name and place to avoid a consideration of this work as local study. We wanted to highlight the methodological approach and the message of the importance of understanding long-term flood frequency/magnitude changes at decadal and centennial scale for improving flood hazard assessments. However, after the reviewer suggestion we agree to include details of the study site in the title as follow:

"Enhanced flood hazard assessment beyond decadal climate cycles based on centennial historical data (Duero Basin, Spain)"

COMMENT: A second suggestion is the possibility to add extreme dates in the title, even if is difficult to fixed the scope, because sometimes are mentioned (and effectively was study) the last 500 years, and in other occasions the period between 1250-1871, maybe because they faced «a non-continuous dataset between 1250 and 1545» (I understand the expression «centennial» in the title).

RESPONSE:

Thank you for this comment. We think that the expression centennial already indicates the use of data records of several hundreds of years. Moreover, the historical temporal framework differs when considering isolated flood data and continuous flood registers. The period between CE 1250 and 1511 has some isolated non-continuous flood data, particularly on extreme events, and therefore, it cannot be included in the flood frequency analysis. Because only reading the paper, the reader will get an idea of the different temporal frameworks for the continuous/discontinuous datasets, we think that is better not to indicate the time framework in the title.

COMMENT:

The authors made a remarkable comparative approach putting its case in a larger frame. I suggest another comparative analysis using an article that tried to estimate the frequency of extraordinary floods of Douro River in the Portuguese territory, till its estuary. (Silva, J.D. da; Oliveira, Manuel de Sousa - *As cheias na parte portuguesa da bacia hidrográfica do rio Douro, ps/p/*. Available https://grupo.us.es/ciberico/.../porto2diasdasilva.pdf.).

RESPONSE:

We have received a pdf copy of the indicated document from the reviewer. A comparative analysis of our results with the one indicated by Silva and Oliveria is out of the scope of this paper, because the important differences in flood response and discharges between the Spanish and Portuguese parts of the Basin. We have added a reference to the indicated study as follow:

Line 584: In Régua the 1739 flood peak was estimated at 18,000 $m^3$/s and in Porto (~105 km downstream) likely reached 20,000 $m^3$/s (Silva and Oliveira, 2002).

COMMENT:It could be important to insist in a comparison between rainfall contrast characteristics consequences (line 181 and further), i.e., the flood peaks contrast between Régua and Porto at the lower basin of Douro, with Valladolid and Zamora and its consequences. The period 1250-1871, in which were identified 69 floods (including ordinary ones), is a number very low if compared with the floods of the Douro in Porto just for the period between 1727 and 1799, in which were found 54 floods (see the quoted article by the authors). Perhaps this increasing number of floods was related to the tidal peaks and the siltation of the estuary, but this contrast of occurrences could open an outlook about long Douro River course behavior, before and after dams' construction.

RESPONSE:

We agree with Prof. Amorim about the strong contrasts on flood peaks between the Portuguese and the Spanish Douro/Duero river and their consequences. It is incredible that some large peaks in Porto were not recorded as large floods in Zamora, but the data shows that the rainfall characteristics are very different. We agree that this topic should be addressed in the future and perhaps an opportunity to collaborate with Portuguese colleagues. Regarding the difference on the number of floods, it is difficult to compare using relative flood classifications, as the ones applied by Alcoforado et al., 2021 and in our paper. In the case of Alcoforado et al., 2021 the frequency of extraordinary floods is

once in each 1.33 years, which is approximately the bankfull discharge of rivers in temperate climates. In the case of Zamora the conditions are drier and probably the historical accounts of high flows without damages are lower than in Porto, where even small floods had important influence in the navegation and port operations. However, the number of large floods are not so different at both sides; in Porto there were six catastrophic floods between 1727 and 1799 (as refered in Alcoforado and colleagues paper) whereas in Zamora we recorded seven floods within relative categories of catastrophic and extraordinary floods in both cases causing moderate to severe damages. As Prof. Amorim knows the tidal conditions were critical in Porto in terms of water stage reached at the lower Douro.

COMMENT:

A final remark: line 575, authors wrote "includes the largest flood on record (Dec 4-6 1739) that reached a stage of 12 m at Dom Luiz I Bridge (Loureiro, 1904; Taborda, 2006)". The bridge doesn't exist in 1739 (only constructed between 1881-1886) but Loureiro (and Taborda quoting Loureiro), used the currently existing bridge as a mark (rebuild the sentence will be enough).

RESPONSE: Thank you very much for this comment, and indeed it is a mistake in the way in which it was indicated the observation site. Indeed, the observation point on stage and flow velocity used by the Porto and Leixôes port authorities directorate is located at a rocky river bank on the right margin, immediately upstream of the D. Luiz bridge, and as the reviewer said was brought to the flood marks as a reference to the Dom Luiz Bridge.

We have modified the sentence as follow:

Lines 581-584: "In the Douro River in Porto these two periods were also identified by their anomalous frequency and severity of floods (Alcoforado et al., 2021; Amorim et al., 2017) that includes the largest flood on record (Dec 4-6, 1739) that reached a stage of 12 m in a bedrock section at the right margin just upstream of the Dom Luiz I Bridge (Loureiro, 1904; Taborda, 2006)."

**REVIEWER COMMENTS by Libor Elleder**

General comments:

The article "Enhanced flood hazard assessment beyond decadal climate cycles based on centennial historical data" presents very a complex study on extreme floods of the Duero River in Zamora. 69 floods including 15 catastrophic ones were identified for time span of 651 years (1250–1871). This count of 15 catastrophic floods represents one catastrophic flood per 41 years on average, which is a realistic assessment. The count of 16 extraordinary floods with discharge maximum over 1900 $m^3 . s^{-1}$ (perception threshold until 1871) were identified. In other words, in1250–1871 this discharge was exceeded on average every 20 years. The largest floods were identified before onset of LIA (1258) and on the end of LIA (1860).

The authors used complex statistical analyses. The flood frequency analyses based on Expected Moments Algorithm (EMA) and Maximum Likelihood Estimator (MLE) methods were combined with five datasets (based on various temporal frameworks). The authors discussed the meteorological framework as well. They present the major floods in context of NAO oscillation. With special interest I have read the chapter on atmospheric rivers influence.

RESPONSE:

We appreciate very much this positive general comments and the interest of the reviewer on the different parts of the paper, moreover knowing the reviewer expertise on quantitative historical Hydrology.

Specific comments

I have noticed a very interesting similarity in condition in old Zamora city and Prague regarding the flood reconstruction.

1/ For both cities, Prague and Zamora, the data gaps occur before the onset of LIA.

2/ For both cities the pictorial documentations were made by (various) Dutch painters (Anton van der Wyngaerde and Egidius Sadeler ) in the 16[th] century.

3/ The hydrological situation was stable from ca 13[th] or 14[th] century to the 1870s.

4/ The perception threshold of *ca* 1900 $m^3$. $s^{-1}$ is similar for period before systematical records.

5/ For Both Zamora and Prague the negative NAO is important.

6/ Some important flood marks were destroyed.

RESPONSE:

We agree on the similarities between Prague and Zamora, and in terms of flood frequency in relation to climate cycles and it shows the influence of the atmospheric circulation changes at European scale. These similarities open an opportunity to analyse the links and connections in terms of seasonal atmospheric patterns.

SPECIFIC COMMENT:

7/ Is the old one bathygraphy of the river channel from 19[th] century in Zamora at disposal?

RESPONSE: No, there is not an old bathymetry of the river channel, only some individual cross-sections at the bridge locations.

8/ Some articles with similar topic should be noticed, discussed and referenced to:

Macdonald, N. 2013. Reassessing flood frequency for the River Trent, Central England, since AD 1320. Hydrology Research 44 (2), 215–233.

Elleder, L., Herget, J., Roggenkamp, T., and Nießen, A.: Historic floods in the city of Prague – reconstruction of peak discharges. Hydrology Research 44, 202–214, 2013.

Wetter, O., Pïster, C., Weingartner, R., Luterbacher, J., Reist, T. & Trösch, J. 2011 The largest floods in the High Rhine basin since 1268 assessed from documentary and instrumental evidence. Hydrol. Sci. J. 56, 733–758.

Aldrete, G. S., 2007. Floods of the Tiber in ancient Rome. Baltimore: John Hopkins University Press 338 pp.. ISBN 0-8018-8405-5

England, J. F. Jr., Jarrett , R. D., Salas, J. D., 2003. Data-based comparisons of moments estimators using historic and paleoflood data. Journal of Hydrology 278, s. 72–196. ISSN 0022-1694

RESPONSE:

Thanks for the suggestions. Some all the suggested references are now included in the manuscript, except the one by England et al., 2003 that we believe it is not needed.

1/ Line 20: AEP is without explanation.

RESPONSE:

Done. Now is included as follow: "The most consistent results were obtained using the HISTO dataset, even for high quantiles (0.001% annual exceedance probability, AEP)."

COMMENT:

2/ Please consider if the "land mark" (first time line 235) expression is suitable: Might be" flood mark" should be more clear.

RESPONSE:

The term "landmark" is more general than flood mark, as it means "a reference point" in our case because it was cited by documents or there may contain flood marks which is a physical gravings or plates made to indicate the flood stage.

We agree that this term may lead to confusion. Therefore, we have change in the case of line 235 to "flood marks" as it is more appropriated in this case. We have gone through the manuscript and change most of the landmarks term by flood marks.

COMMENT:

3/ The authors use 2 categories of floods, i.e. catastrophic and extraordinary. Please, use it also in Table 1.

RESPONSE:

The descriptor of the flood category was now added in Table 1.

4/ Line 580 LIE. What does it mean?

RESPONSE:

Thank you to note the mistake. It should say LIA, that it is an abbreviation for Little Ice Age. It was changed and also in the previous sentence was written in full as follow:

"In terms of flood magnitude, the Duero data suggest that the largest floods occurred at the onset and final stages of the Little Ice Age (LIA)."

5/ Please, check the figures and enlarge the letters to make the figures better readable.

RESPONSE:

Thanks. We will enlarge the letters on the figures for the final submission.

7/ Abbreviation FFA (flood freq. an.) is explained on line 630, but mentioned earlier for the first time. Please, revise.

RESPONSE:

Thanks for alert on this problem. It was now added in full in the first sentence that appears in the text (line 506)